# TAD boundary deletion causes *PITX2*-related cardiac electrical and structural defects

Manon Baudic [1,28], Hiroshige Murata [2,28], Fernanda M. Bosada [3,28], Uirá Souto Melo [4,28], Takanori Aizawa [5,28], Pierre Lindenbaum [1,28], Lieve E. van der Maarel [6,28], Amaury Guedon[1], Estelle Baron[1], Enora Fremy[1], Adrien Foucal[1], Taisuke Ishikawa [7], Hiroya Ushinohama[8], Sean J. Jurgens [9,10], Seung Hoan Choi [9,11], Florence Kyndt[1], Solena Le Scouarnec[1], Vincent Wakker[3], Aurélie Thollet[1], Annabelle Rajalu[1], Tadashi Takaki[12,13,14], Seiko Ohno [15], Wataru Shimizu[2], Minoru Horie [16], Takeshi Kimura[5], Patrick T. Ellinor [9,17,18], Florence Petit [19,20], Yves Dulac[21], Paul Bru[22], Anne Boland[23], Jean-François Deleuze[23], Richard Redon [1], Hervé Le Marec [1], Thierry Le Tourneau [1], Jean-Baptiste Gourraud[1,24], Yoshinori Yoshida [12], Naomasa Makita [7,25], Claude Vieyres[26], Takeru Makiyama[5,27,29], Stephan Mundlos[4,29], Vincent M. Christoffels [6,29], Vincent Probst[1,24,29], Jean-Jacques Schott[1,24,29] ✉ & Julien Barc [1,24,29] ✉

While 3D chromatin organization in topologically associating domains (TADs) and loops mediating regulatory element-promoter interactions is crucial for tissue-specific gene regulation, the extent of their involvement in human Mendelian disease is largely unknown. Here, we identify 7 families presenting a new cardiac entity associated with a heterozygous deletion of 2 CTCF binding sites on 4q25, inducing TAD fusion and chromatin conformation remodeling. The CTCF binding sites are located in a gene desert at 1 Mb from the *Paired-like homeodomain transcription factor 2* gene (*PITX2*). By introducing the ortholog of the human deletion in the mouse genome, we recapitulate the patient phenotype and characterize an opposite dysregulation of *PITX2* expression in the sinoatrial node (ectopic activation) and ventricle (reduction), respectively. Chromatin conformation assay performed in human induced pluripotent stem cell-derived cardiomyocytes harboring the minimal deletion identified in family#1 reveals a conformation remodeling and fusion of TADs. We conclude that TAD remodeling mediated by deletion of CTCF binding sites causes a new autosomal dominant Mendelian cardiac disorder.

Whole genome sequencing allows to interrogate the non-coding portions of the genome and offers the possibility to provide a molecular diagnosis to patients with unsolved genetic disorders[1]. The challenge resides in the translation of the identified non-coding genetic variants to phenotype expression, since the functional effect of such variation remains hard to predict. The non-coding genome harbors regulatory sequences such as enhancers that can activate gene transcription from great distances and sequences involved in chromatin structure. Regulatory sequences and their target genes are often confined to shared topologically associating domains (TADs)[2,3]. Regulatory regions are brought in close proximity to target promoters by chromatin loops organized in TADs[4]. The formation and maintenance of loops and

TADs involves the interplay of the cohesin complex and the DNA-binding protein CTCF, which bring distal sequences together[5]. Genome editing of TAD boundaries in mice leads to aberrant gene regulation and altered phenotypes[6]. In the mammalian genome, CTCF is dose-dependently required for looping between CTCF target sites and insulation of topologically associating domains (TADs)[7]. In a mouse model, cardiomyocyte-specific inducible depletion of CTCF caused profound transcriptional dysregulation and heart failure[8]. These studies demonstrate the potential pathogenicity of genetic variants affecting CTCF sites at TAD boundaries.

Here, we describe 7 families that share overlapping deletions in a 1.5 Mbp gene desert on chromosome 4q25 presenting with a novel cardiac entity including sinoatrial node (SAN) dysfunction, atrial fibrillation and developmental defects. The smallest deletion harbors two diverging CTCF binding sites and disrupts a TAD boundary. Among the genes located in the TADs, PITX2, a homeobox transcription factor, plays a key role in left-right asymmetrical development of organs including gut, stomach and heart, in cell fate determination, differentiation and organogenesis[9]. Moreover, PITX2 is involved in the regulation of genes underlying electrophysiological properties of the left atrium[10,11] and the *PITX2* locus has been implicated in atrial fibrillation by genome-wide association studies[12]. *Pitx2* insufficiency in mice led to arrhythmogenesis and ectopic activation of aspects of the SAN genetic program in left atria[13]. Using human induced pluripotent stem cell (hiPSC) models and mouse models, we show that the deletions in the families cause chromatin conformation changes and dysregulation of *PITX2* expression in the heart, explaining the observed phenotypes.

## Results

### Intergenic deletion associated with a novel cardiac entity

By linkage analysis, we have previously reported a haplotype of 17 Mb at the 4q25 locus co-segregating perfectly with a complex cardiac electrical and structural phenotype in a large French family[14]. Whole-exome sequencing failed to identify any pathogenic coding variant within this interval. By whole genome sequencing (WGS), we then identified a heterozygous 15 kb deletion in an intergenic segment at this locus (4q25-DEL-15Kb, chr4:112555060-112570371, *GRCh37−hg19*), absent from the public databases gnomAD[15] and Database of Genomic Variants[16] (Fig. 1a, Family#1). The closest gene upstream of the deletion, *PITX2*, is located 1 Mb away while the nearest gene downstream−*FAM241a*−is at a distance of 500 kb from the deletion (Fig. 1b). We then investigated additional unrelated familial cases with similar cardiac phenotypes originating from France and Japan, and identified six additional deletions of variable sizes at 4q25 overlapping the first 15 kb deletion (Fig. 1a, b).

Phenotype severity or complexity was comparable among the seven families despite deletion sizes ranging from 15 kb to 330 kb (Supplementary Fig. 1a). All affected probands and relatives presented with electrical cardiac disorder and/or cardiac structural abnormalities at young age (Table 1 and Supplementary Data 1). Sinus node dysfunction, which is characterized by sinus bradycardia with sometimes cardiac pauses, was the main phenotype for 94% of the deletion carriers ($n = 65/69$, mean age: 13.3 y.o. ± 14.4), of whom 60% benefited from pacemaker implantation ($n = 39/65$, mean age: 28 y.o. ± 17.6) (Fig. 1c, Supplementary Data 1 and Supplementary Fig. 1a). No sinus node dysfunction was identified among non-carriers ($p < 2.2E−16$) (Fig. 1c). Heart rate measured by the RR interval (duration between two heart beats) was significantly longer in deletion carriers compared to non-carriers ($1037 ± 315$ vs. $758 ± 184$ ms, $p < 1.0E−04$). Of note, 45% of carriers ($n = 26/58$) presented atrial fibrillation (mean age: 40 y.o. ± 14.7) vs. 3% of the non-carriers ($p = 2.1E−05$) (Table 1 and Fig. 1c). The ventricular repolarization duration (QTc), while remaining within the normal limits, was longer among subjects carrying the deletion than among non-carriers (mean QTc: $421 ± 43$ vs. $397 ± 30$ ms, $p = 0.02$). Similarly, the ventricular repolarization duration including

the ECG "U" wave QTUc was significantly prolonged in deletion carriers compared to non-carriers ($575 ± 79$ vs. $441 ± 51$ ms, $p < 1.0E−04$). Cardiac morphological abnormalities were found in 72% ($n = 42/58$) of the clinically explored deletion carriers, whereas they were found in only 14% ($n = 4/28$) of the individuals carrying no deletion ($p = 4.6E−07$). These abnormalities included mitral valve prolapse/billowing (38% of deletion carriers vs. 15% in non-carriers; $p = 4.3E−02$), left-ventricular non-compaction (24% vs. 0%; $p = 3.8E−03$) and atrial septal defect (24% vs. 0% $p = 3.7E−03$) (Table 1, Fig. 1c and Supplementary Fig. 1b). Together, these electrical anomalies associated with cardiac structural defects constitute a new cardiac entity with a high penetrance (97%, $n = 67/69$) (Table 1 and Supplementary Data 1).

### Orthologous deletion in mice recapitulates the human phenotype

To investigate the causal relationship between the 4q25 deletions and this complex cardiac phenotype, we generated a homozygous knockout mouse model for the orthologous 15 kb deletion identified in Family#1, hereafter called *DelB* (Fig. 1d). We observed a longer RR interval among *DelB* mice compared to WT mice (181.7 ms vs. 133.9 ms respectively; $p = 2E−4$) as well as higher heart rate variation ($21 ± 5.8$ ms, $3.1 ± 0.2$ ms; $p = 6.3E−5$), sinus node dysfunction with longer heart rate-corrected sinus node recovery time ($169 ± 17$ ms, $53 ± 5.1$ ms; $p = 3.4E−6$) and longer Wenckebach cycle length ($87 ± 2.2$ ms, $80 ± 0.9$ ms; $p = 8.9E−3$) (Fig. 1e and Supplementary Fig. 2a). Episodes of atrial arrhythmias were more frequent (73% [11/15] vs. 20% [5/25]; $p = 2.1E−3$) and lasted longer ($134 ± 56$ s, $1.6 ± 0.9$ s; $p = 1.2E−5$) in *DelB* mice (Fig. 1e and Supplementary Fig. 2b), which also presented atrial septal defect (42% in *DelB* [5/12] vs. 0 in WT [0/10]; $p = 3.9E−2$) (Fig. 1e and Supplementary Fig. 2c) as well as larger heart in comparison to WT mice (Heart Weight/Body Weight: $0.55 ± 0.009\%$ in WT vs. $0.58 ± 0.011\%$ in DelB; $p = 2.6E−2$) (Supplementary Fig. 2d). Altogether, these findings demonstrate the functional role of this genomic interval in cardiac electrical activity and heart development, and the causal role of this deletion in the observed cardiac disorder in human.

### The deletion of CTCF binding sites induce 3D chromatin remodeling

To decipher possible molecular mechanisms by which the intergenic deletion at 4q25 affects the transcriptional output of the locus, we investigated the epigenetic landscape in WT human cardiomyocytes derived from induced pluripotent stem cells (hiPSC-CM WT). Chromatin accessibility, measured by ATAC-seq, revealed a unique open chromatin region (-350 bp) located within the 15 kb deleted region (Fig. 2), also identifiable in adult human ventricular cardiomyocytes from the Cardiac Atlas Regulatory Element (CARE) datasets[17]. Histone modifications associated with promoters (H3K4me3), enhancers (H3K4me1) and active state (H3K27ac) were not observed at this location in hiPSC-CM (Fig. 2). However, we identified 2 diverging binding sites (Supplementary Fig. 3) for the CCCTC-binding factor (CTCF) known to be involved in the formation of loops and TAD boundaries. In mouse, the conserved CTCF sites interact with a CTCF site in *Pitx2* in a CTCF-dependent manner, together marking the boundaries of a TAD[18]. On a chromatin contact map from hiPSC-CM WT, we observed that the CTCF sites are located at a TAD boundary, separating the TAD containing *PITX2* (hereafter named *PITX2*-TAD) from a TAD without any annotated coding gene (non-coding-TAD) (Fig. 2). The *PITX2*-TAD harbors only *PITX2* as a protein-coding gene and annotated non-coding RNA. The non-coding-TAD harbors only poorly annotated long non-coding RNAs. We then investigated the impact of the minimal deletion on chromatin topology in human cardiomyocytes by generating an isogenic cell line derived from hiPSC WT containing the homozygous 15 kb deletion (hiPSC 15 kb −/−). By Hi-C, we observed that *PITX2*-TAD and non-coding-TAD were fused,

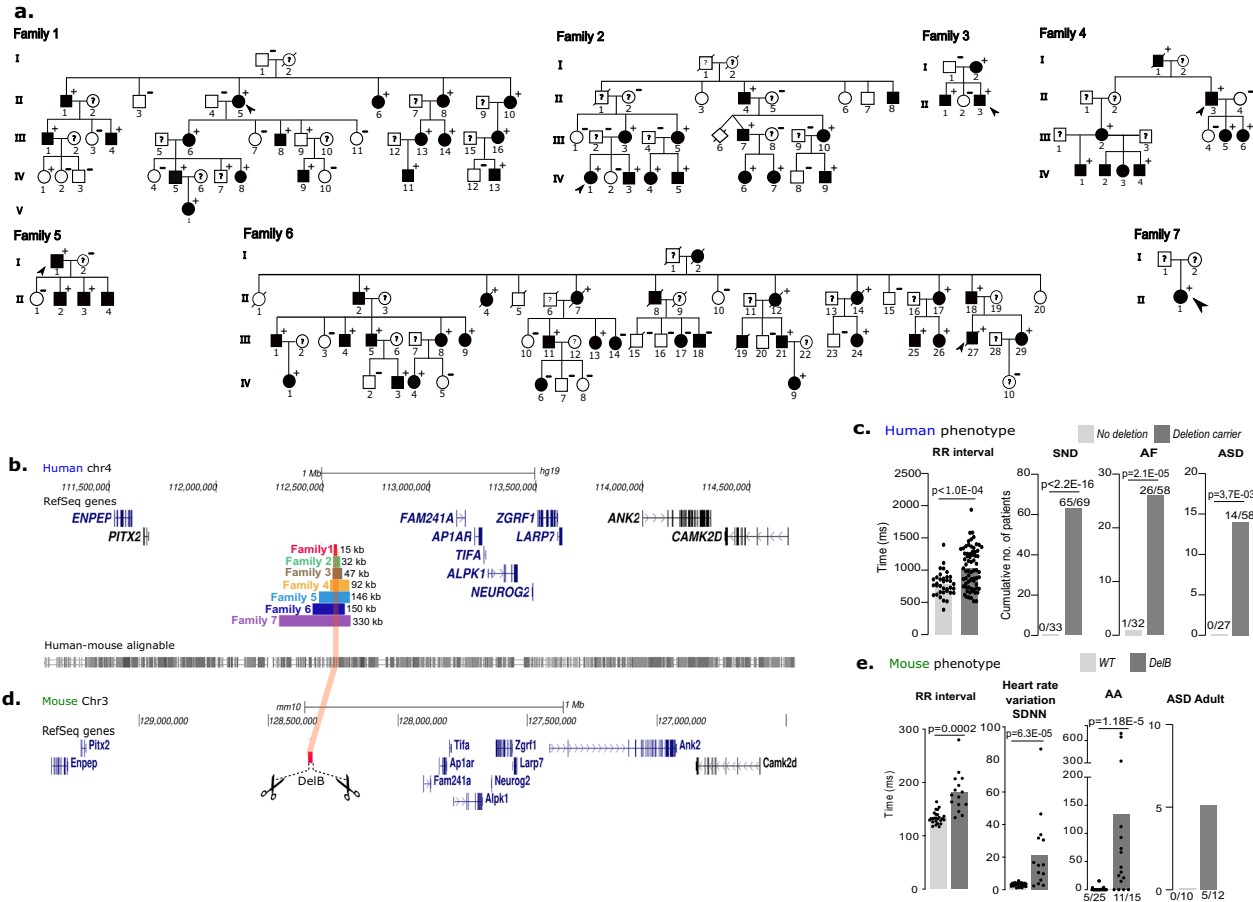

**Fig. 1 | Overlapping deletion in seven families presenting a new cardiac entity associating electrical disorder and developmental defects recapitulated in a mouse model. a** Pedigree of seven families presenting a new cardiac entity. Symbols marked by a slash indicate that the subject was deceased. Males were indicated by squares and females by circles. Black symbols represent individuals who present at least one of the cardiac phenotypes listed in Table 1 according to clinical examination. "+" symbol denotes deletion carriers and "−" symbol non-carrier. The proband is indicated by the arrow. **b** Overview of the 4q25 human locus in UCSC Genome browser with the reference genome assembly *hg19*. Genes are represented in black and blue. 7 Deletions sizes are represented in different horizontal color lines: red/Family#1 (chr4:112555060-112570371), green/Family#2, brown/Family#3, yellow/Family#4, Blue/Family#5, dark blue/Family#6 and purple/Family#7. Human-mouse conservation is measured by Cons 46-way (*phastCons* and *phyloP* score) provided by UCSC Genome browser. **c** Graphs depict individual and average measurements for RR interval and cumulative number of patients presenting sinus node dysfunction (SND), atrial fibrillation (AF), atrial septal defect (ASD) in deletion carriers (RR interval *n* = 70, SND *n* = 69, AF *n* = 58, ASD *n* = 58) and non-carriers (RR

interval *n* = 31, SND *n* = 33, AF *n* = 32, ASD *n* = 27) in the seven families. Significance for each parameter was determined with two-sided Mann–Whitney test (RR interval *p* < 1.0E−04) and two-sided Fisher's exact test (SND *p* < 2.2E−16, AF *p* = 2.1E−05, ASD *p* = 3.7E−03). **d** Representation of the reverse orthologous mouse locus in UCSC Genome browser with the reference genome assembly *mm10*. Genes are represented in black and blue. Homozygous 15Kb deletion generated in the mouse model is represented in red delimited by the dashed black lines. **e** Graphs show individual and average RR interval, Standard deviation of NN intervals (SDNN; heart rate variation), the duration of atrial arrhythmia (AA) episode (*WT*, *n* = 25, *DelB*, *n* = 15) after two pacing passes, number of mice with at least one episode lasting >1 s is indicated above each bar graph of control (*n* = 25) and *DelB* (*n* = 15) mice. Number of mice from control (*n* = 10) and *DelB* (*n* = 12) presenting atrial septal defect are depicted above bar graph. Significance for each parameter was determined with two-sided Welch's *t* test (RR *p* = 0.0002), two-sided Mann–Whitney test (SDNN *p* = 6.3E−05, AA duration *p* = 1.18E−5), or two-sided Fisher's exact test (AA inducibility *p* = 0.0021, ASD Adult *p* = 0.0396). Individual data are available in Supplementary Data 1 and in the Source Data file.

reshaping the 3D chromatin structure and changing interactions at this locus (Fig. 3a and Supplementary Fig. 4). Of note, the downstream TADs containing coding genes (*TIFA, FAM241A, AP1AR, LARP7, ANK2 and CAMK2D*) remained unchanged (Supplementary Fig. 4a).

### *Pitx2* expression is deregulated in *DelB* mouse hearts

We investigated the impact of the deletion on the expression level of genes located at 1 Mb on both sides of the deletion in the left atria, right atria and ventricles of mice (Fig. 4a). We found a marked induction of expression of *Pitx2* (*p* = 0.0009) as well as moderately decreased expression of *Fam241a* and *Tifa* (*p* = 0.045 and *p* = 0.0007) in the right atria of *DelB* mice. Only the expression of *Tifa* varied slightly in the left atria (*p* = 0.04) while expression of *Enpep, Fam241a, Ank2* and *Pitx2* was slightly decreased in ventricles (*p* < 0.0001,

*p* < 0.0001, *p* = 0.0001 and *p* = 0.0007, respectively). We generated two additional transgenic mouse models carrying orthologous deletions including respectively the upstream (*DelA*) and downstream (*DelC*) intervals deleted in Family#4 (Supplementary Fig. 5a) and observed no differential expression in *DelA* mice and slight changes in the expression levels of *Camk2d, Ap1ar* and *Ank2* in *DelC* mice (Supplementary Fig. 5b).

We separately microdissected SAN tissue and RA tissue without SAN of adult WT (*n* = 8) and *DelB* (*n* = 7) mice and performed whole-tissue RNA sequencing (Fig. 4b–d). We observed that *Pitx2* was significantly induced only in *DelB* SAN tissue (1.04 L2FC, *p*adj = 1.26E−04) and not in RA tissue (0.07 L2FC, *p*adj = 0.58). These data reveal that *Pitx2* is specifically induced in SAN tissue of *DelB* mice. In SAN tissue, we found 2182 differentially expressed genes (*p*adj < 0.05; Fig. 4c and

**Table 1 | Genotype-phenotype correlation between deletion carriers and non-deletion carriers**

| Total population | Deletion carriers n = 70 | | | | Deletion non-carriers n = 36 | | | | p value |
| | Females = 36, Males = 34 | | | | Females = 22, Males = 14 | | | | 0.4 |
| | | Number of individuals | | | | Number of individuals | | | |
| | | Yes | No | NA | | Yes | No | NA | |
|---|---|---|---|---|---|---|---|---|---|
| Cardiac electrical disorders | 97% | 67 | 2 | 1 | 3% | 1 | 31 | 10 | <2.2E−16 |
| Sinus node dysfunction | 94% | 65 | 4 | 1 | 0% | 0 | 33 | 3 | <2.2E−16 |
| Pacemaker | 57% | 39 | 30 | 1 | 0% | 0 | 32 | 4 | 1.7E−09 |
| Atrial fibrillation | 45% | 26 | 32 | 12 | 3% | 1 | 31 | 4 | 2.1E−05 |
| RR interval (SD) | 1037 ms (315) | 70 | / | 0 | 758 ms (184) | 31 | / | 5 | <1.0E−04 |
| QTc Bazett (SD) | 421 ms (43) | 70 | / | 0 | 397 ms (30) | 32 | / | 4 | 0.02 |
| QTUc (SD) | 575 ms (79) | 60 | / | 10 | 441 ms (51) | 19 | / | 17 | <1.0E−04 |
| Cardiac structural defects | 72% | 42 | 16 | 12 | 14% | 4 | 24 | 14 | 4.6E−07 |
| Atrial septal defect | 24% | 14 | 44 | 12 | 0% | 0 | 27 | 9 | 3.7E−03 |
| Non-compaction of the left ventricle | 24% | 13 | 42 | 15 | 0% | 0 | 27 | 9 | 3.8E−03 |
| Mitral valve prolapse or billowing | 38% | 21 | 35 | 14 | 15% | 4 | 23 | 9 | 4.3E−02 |
| Vena cava azygos return | 9% | 5 | 49 | 16 | 0% | 0 | 27 | 9 | 0.2 |
| Pulmonary valve stenosis | 7% | 4 | 51 | 15 | 0% | 0 | 27 | 9 | 0.3 |

Significance p value for each parameter was determined with two-sided Fisher's exact test (Symptomatic, sinus node dysfunction, Pacemaker, Atrial Fibrillation, Holter ventricular hyperexcitability, Ventricular arrhythmia after PM implantation, Mitral valve prolapse or billowing, Atrial septal defect, Non-compaction of the left ventricle, Vena cava azygos return, Pulmonary valve stenosis), two-sided Mann–Whitney test (QTc Bazett) and two-sided Welch's t test (QTUc). Individual data are available in Supplementary Data 1.

Supplementary Data 2) of which 1003 were upregulated and 1179 were downregulated in *DelB* mice. We additionally found that the expression of 13 genes located within 3 Mb around the deletion was unchanged, with the exception of *Larp7*, an RNA binding protein recently implicated in mitochondrial homeostasis and the maintenance of cardiac function[19], which was slightly upregulated in *DelB* SAN tissue (Fig. 4b and Supplementary Data 2). Gene Ontology (GO) analysis yielded terms for neuronal and development processes for genes downregulated in the *DelB* SAN and terms for metabolic processes for genes upregulated in the *DelB* SAN (Supplementary Data 3 and 4). We additionally noted that the ectopic expression of *Pitx2* in the SAN region was accompanied by a marked reduction in pacemaker cell-associated gene expression in *DelB* SAN tissue, including key transcription factor genes (*Tbx3, Isl1*), ion channels (*Hcn4, Cacna2d2, Ryr2*) and other SAN markers (*Vsnl1, Ntm, Cpne5*)[20–24]. Tbx3 and Isl1 are essential for SAN development[20,25,26] and *Hcn4* encodes the hyperpolarization-activated cyclic nucleotide-gated $K^+$ channel that mediates the spontaneous activation of pacemaker cells in the SAN and has been implicated in SND[27]. To further explore the changes in pacemaker/SAN-enriched gene expression in our whole-tissue RNA-seq data, we compared the genes differentially expressed in the *DelB* SAN with those previously found by single cell RNA-sequencing to be enriched or depleted in fetal pacemaker cardiomyocytes compared to atrial cardiomyocytes[22]. Of 794 pacemaker enriched genes, 93 were downregulated in *DelB* SAN tissue, including all well-established pacemaker marker genes, and 106 were upregulated (Fig. 4d and Supplementary Data 5). Of 497 pacemaker depleted genes, 69 were downregulated in *DelB* SAN tissue, and 116 upregulated, including atrium-specific markers *Nppb* and *Bmp10* (Fig. 4d and Supplementary Data 3). GO analysis for transcripts enriched in pacemaker cardiomyocytes and upregulated in the *DelB* SAN yielded terms including oxidative phosphorylation, while that of transcripts down-regulated in the *DelB* SAN yielded terms involved in ion handling/transport (Supplementary Data 6–9). Conversely, GO analysis for transcripts depleted in pacemaker cardiomyocytes and upregulated in the *DelB* SAN yielded terms including translation, and that of genes downregulated in the *DelB* SAN included terms associated with morphogenesis. These findings suggest that the ectopic expression of *Pitx2* induces a partial phenotypic transformation toward an atrial working

myocardial phenotype in the SAN region, coinciding with SAN dysfunction and atrial arrhythmias in adult *DelB* mice. Altogether, our results indicate that the orthologous 15 kb deletion (*DelB*) modulates *Pitx2* expression in the SAN and ventricle, respectively, which likely mediates the cardiac electrical and structural phenotype observed in patients.

## PITX2 expression is differentially dysregulated in atrial and ventricular like human cells

In order to assess a causal relationship between the deletion of the CTCF-binding sites in patients and cardiac transcriptional remodeling, we generated a cell line isogenic to hiPSC WT in which the CTCF binding sites were homozygously deleted (hiPSC-CM CTCF−/−). Remarkably, we observed reduced expression of *PITX2* in ventricular like cells derived from hiPSC-CM CTCF−/− (Fig. 4e), and increased *PITX2* expression in pacemaker-like cells (Fig. 4e), similar to the changes of *Pitx2* expression observed in *DelB* mice. The expression of the lnRNAs located in the 4q25 region (*PANCR, LINC01438, MIR297, RNU6-289P, AC13918.1, AC0139718.2 and LINCO2945*) in the ventricular like hiPSC-CM WT and CTCF−/− was considered to be at background level (Supplementary Fig. 6). These results from hiPSC-CM recapitulate the *Pitx2* dysregulation identified in vivo in the *DelB* mouse model and confirm the likely molecular mechanism underlying this new cardiac entity in patients.

## New interactions between atrial and ventricular-specific regulatory regions and PITX2

To explore the compartment-specific deregulation of *PITX2* in deletion carriers, we investigated the potential novel interactions and regulation of *PITX2* resulting from the TAD fusion. Subtraction Hi-C maps showed a loss of interactions within the original *PITX2*-TAD while a gain of interactions was observed between *PITX2*-TAD and the non-coding-TAD region (Fig. 3a). In the region gaining interaction with *PITX2*, chromatin accessibility maps in human adult atrial and ventricular tissues[17] point to two non-coding regions (R1 and R2) that show differentially accessible regions between atrial and ventricular tissues (Fig. 3b). These two regions harbor H3K27ac and H3K4me1 histone marks in hiPSC-CM WT compatible with active enhancer regions (Fig. 3b). Interestingly, R1 and R2 harbor hiPSC-derived pacemaker cardiomyocyte-specific accessible chromatin regions (Fig. 3b)[28].

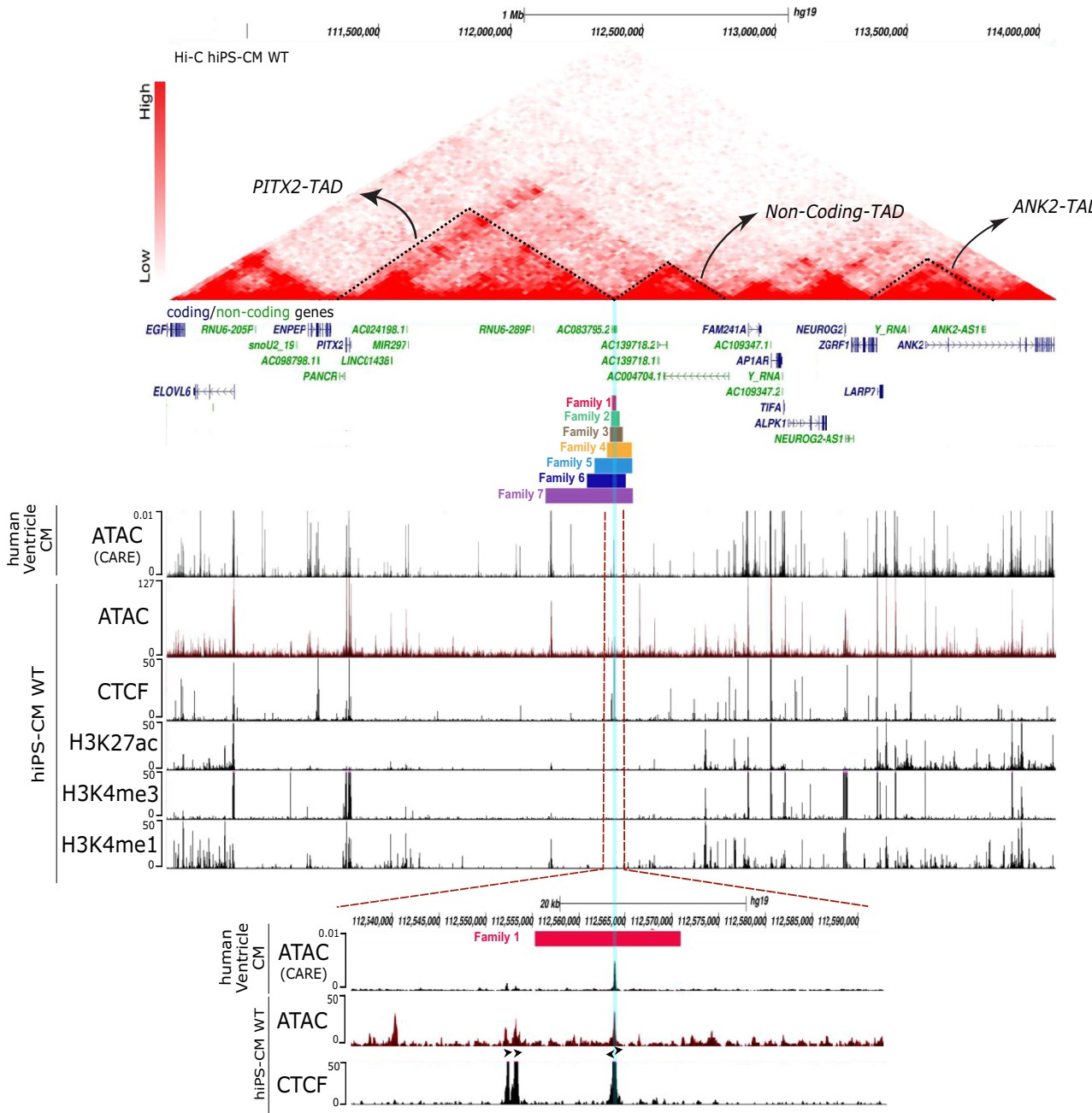

**Fig. 2 | The 15Kb region contains CTCF TAD boundary.** Annotation of the 4q25 region with Hi-C map interaction profile from hiPSC-CM WT. Dashed triangles delimit the PITX2-TAD, Non-coding-TAD, ANK2-TAD. Coding genes are represented in blue and non-coding genes in green and 7 Deletions sizes are represented in different horizontal color lines: red/Family#1 (chr4:112555060-112570371), green/Family#2, brown/Family#3, yellow/Family#4, Blue/Family#5, dark blue/Family#6 and purple/Family#7. ATAC-seq tracks from adult human ventricular tissue from CARE database, ATAC-seq from hiPSC-CM WT, CUT&RUN tracks from hiPSC-CM WT using H3K27ac, H3K4me3 and H3K4me1 and CTCF antibodies are displayed. Zoom in of the 15Kb region annotated with ATAC and CUT&RUN CTCF tracks are represented in the bottom panel. The blue line highlights the open region containing the CTCF binding sites. Black arrows represent CTCF binding site orientation. Genome tracks were generated using UCSC Genome browser with the reference genome assembly *hg19*.

Furthermore, binding sites for cardiac transcription factors[29–31] such as ISL1, TBX5, GATA4, NKX2-5, MEIS1, TEAD and TBX5 co-colocalize with these accessible regions, supporting their potential role in cardiac gene expression regulation (Fig. 3b).

Interestingly, R1 corresponds to the mouse orthologue lncRNA *D030025E07Rik*, also known as *Playrr* (Fig. 3a). In the developing dorsal mesentery, *Playrr* and *Pitx2* are expressed in a mutually exclusive manner and their reciprocal repression has been proposed to contribute to the establishment of left-right asymmetry during gut looping[18]. Mouse *Playrr* is localized in the non-coding TAD. We

interrogated SAN and RA-specific RNA-seq data sets[20] and found that *Playrr* is expressed in the early fetal SAN and to a lesser extent in the RA (*p* = 0.02) (Fig. 5a). Just before birth, the levels of *Playrr* expression were similar in SAN and RA and after birth, *Playrr* expression was decreased in both tissues (Fig. 5b). Although *PLAYRR* has not been annotated in the human genome, we observed SAN-enriched expression of a human orthologous transcript (*PLAYRR*) in RNA-seq data from fetal human SAN and RA[21] (Fig. 5c). In both species, the transcript is initiated from a conserved promoter and appears to be poorly spliced. RNA sequencing of the adult micro-dissected SANs reveals a marked

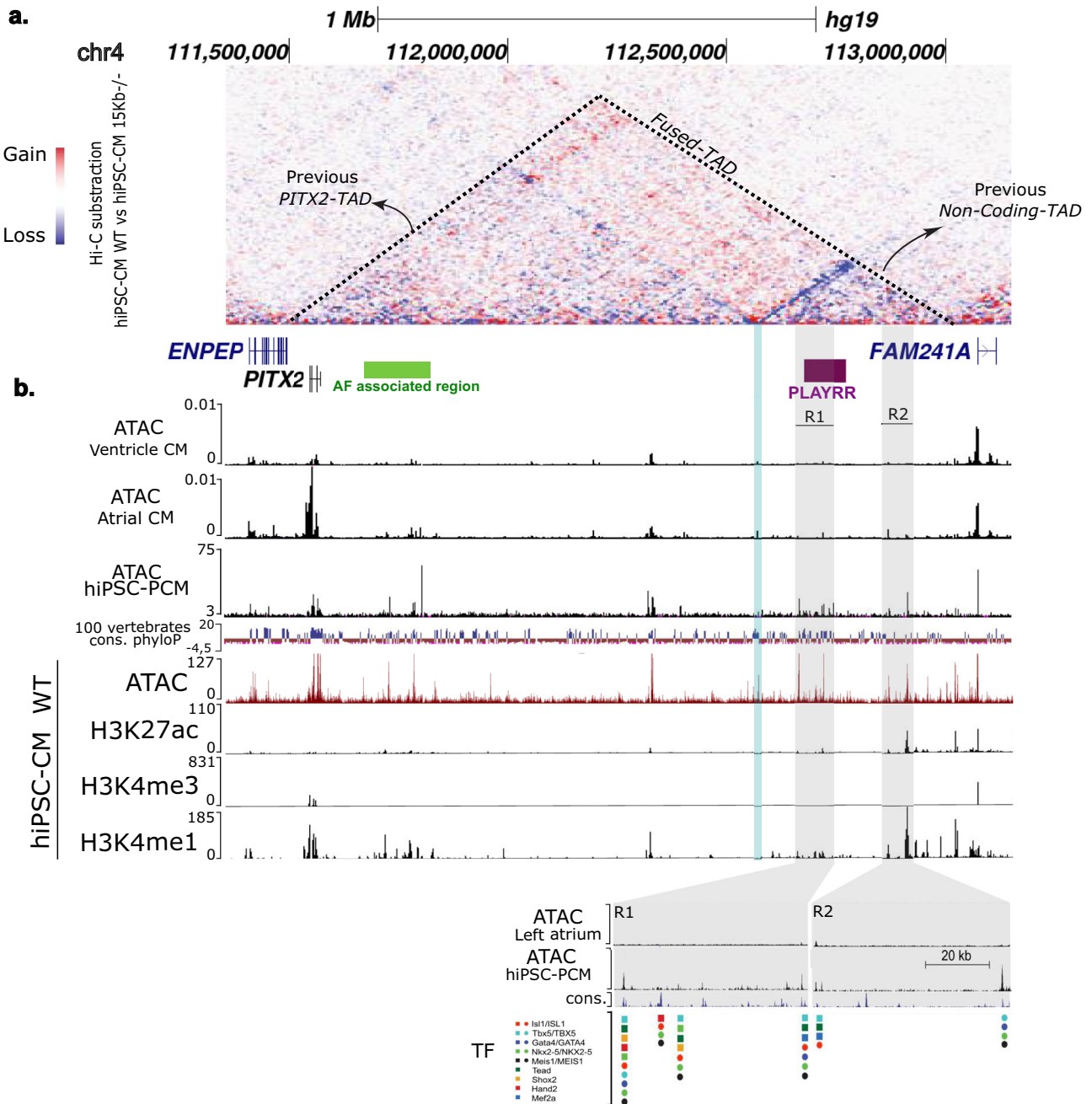

**Fig. 3 | 3D chromatin remodeling induces new interaction between *PITX2* and SAN specific regulatory region resulting to *PITX2* expression remodeling. a** Hi-C subtraction map of hiPSC-CM WT and hiPSC-CM 15Kb−/− (red indicates gain of interaction and blue indicates loss of interaction). Dashed triangles delimit the fused TAD. Coding genes are represented in blue and non-coding genes in green. **b** Annotation of the 4q25 region with ATAC-seq from adult atrial and ventricular tissues from CARE database (*hg38*) and from hiPSC-derived pacemaker-like cardiomyocytes. Genome conservation across 100 vertebrates is presenting with phyloP score. Location of AF associated region in the genome is highlight with a green horizontal line and location of *PLAYRR* in purple horizontal line. The Blue line highlights the open region containing the CTCF binding sites and gray lines highlight regulatory regions newly interacting with *PITX2* (R1 and R2). Annotation of the 4q25 region with, ATAC-seq from hiPSC-CM WT, CUT&RUN tracks from hiPSC-CM WT using H3K27ac, H3K4me3 and H3K4me1. Annotation of R1 and R2 regulatory regions with human and mouse binding sites for cardiac transcription factors (ISL1, GATA4, NKX2-5, MEIS1, TEAD and TBX5). Genome tracks were generated using UCSC Genome browser with the reference genome assembly *hg19*.

reduction in *Playrr* expression in *DelB* mice as compared to WT mice (L2FC = −2.45, *p* = 0.007), complementing the SAN-specific gain in *Pitx2* expression (L2FC = 2.14, *p* = 0.0002) (Fig. 5d).

## Discussion

Chromatin topology plays an essential role in gene regulation, development and disease. The discovery of TADs, domains of preferential interaction topologically separated from adjacent domains, has

provided insight into the mechanistic relation between regulatory sequences and genes within complex regulatory landscapes[32]. Investigation of mutations inducing TAD remodeling as a molecular mechanism underlying Mendelian diseases is highly amenable by whole genome sequencing screening combined with comprehensive functional genomic annotation[33,34]. Here, we describe a new cardiac entity associating cardiac electrical and structural defects that are likely mediated by heart compartment-specific dysregulation of *PITX2*

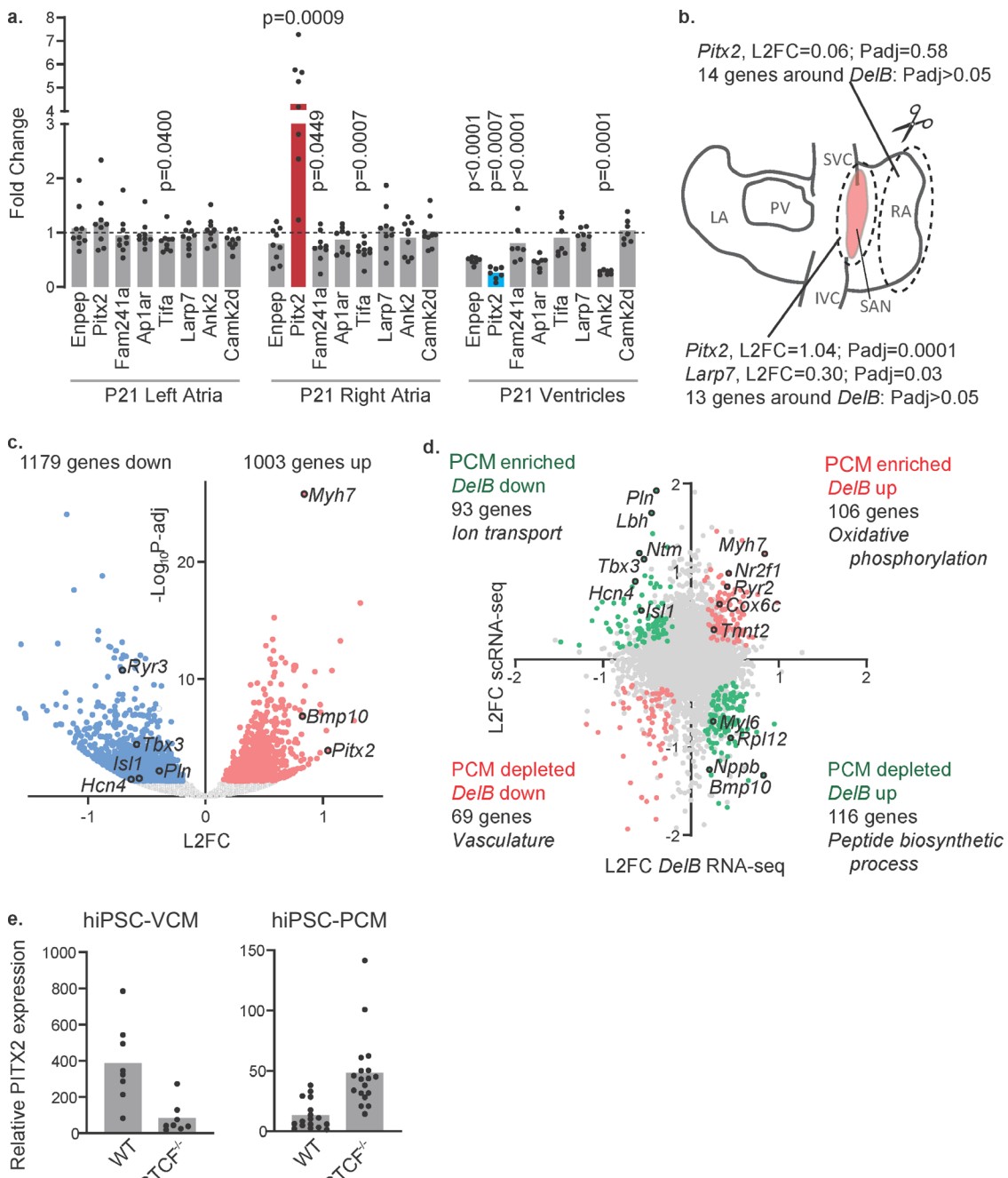

**Fig. 4 | Pitx2 expression is deregulated in the *DelB* SAN and ventricles. a** Fold change expression levels of detectable TAD genes in P21 *DelB* ($n = 9$) vs. control littermates (dotted line) in the left and right atria and ventricles. Red and blue bars highlight upregulation ($p = 0.0009$) and downregulation ($p = 0.0007$) of Pitx2 respectively. Statistical significance was determined across four genotypes and within each tissue type using Kruskal–Wallis test followed by pairwise comparisons with Dunn's multiple comparison tests in right and left atrial, and Welch's ANOVA followed by Dunnett's T3 multiple comparison tests in ventricles. Significant *p* values adjusted for multiple testing are shown in the graph. **b** Whole-tissue RNA sequencing was performed on micro-dissected SAN (in red) and RA regions (PCM pacemaker cardiomyocyte, LA left atrium, PV pulmonary vein, SVC superior caval vein, IVC inferior caval vein, RA right atrium, SAN sinoatrial node). Expression analysis of microdissected SANs of wild type and *DelB* mice show that Pitx2 and Larp7 are the only differentially expressed TAD genes between control ($n = 8$) and

*DelB* ($n = 7$) adult SANs. Differential expression analysis was performed using the DESeq2 package. *p* values were corrected for multiple testing using a false discovery rate with 0.05. **c** Expression analysis of microdissected SANs of wild type and *DelB* mice show differential expression of genes between control and *DelB* adult mice. Blue dots highlight genes downregulated and red dots highlight genes upregulated in *DelB* mice. The differential expression analysis was performed using the DESeq2 package with a two-sided Wald test and multiple testing correction using a false discovery rate with 0.05. **d** Scatter plot showing the relative expression of genes enriched in the pacemaker cardiomyocyte cluster alongside the differential expression of genes between control ($n = 8$) and *DelB* ($n = 7$) SAN region. **e** Relative *PITX2* expression in hiPSC-CM WT and hiPSC-CM-CTCF−/− ventricle like cardiomyocytes (hiPSC-VCM $n = 8$, $n = 8$) and pacemaker cell like cardiomyocytes (hiPSC-PCM $n = 13$, $n = 19$). Source data are provided as a Source Data file.

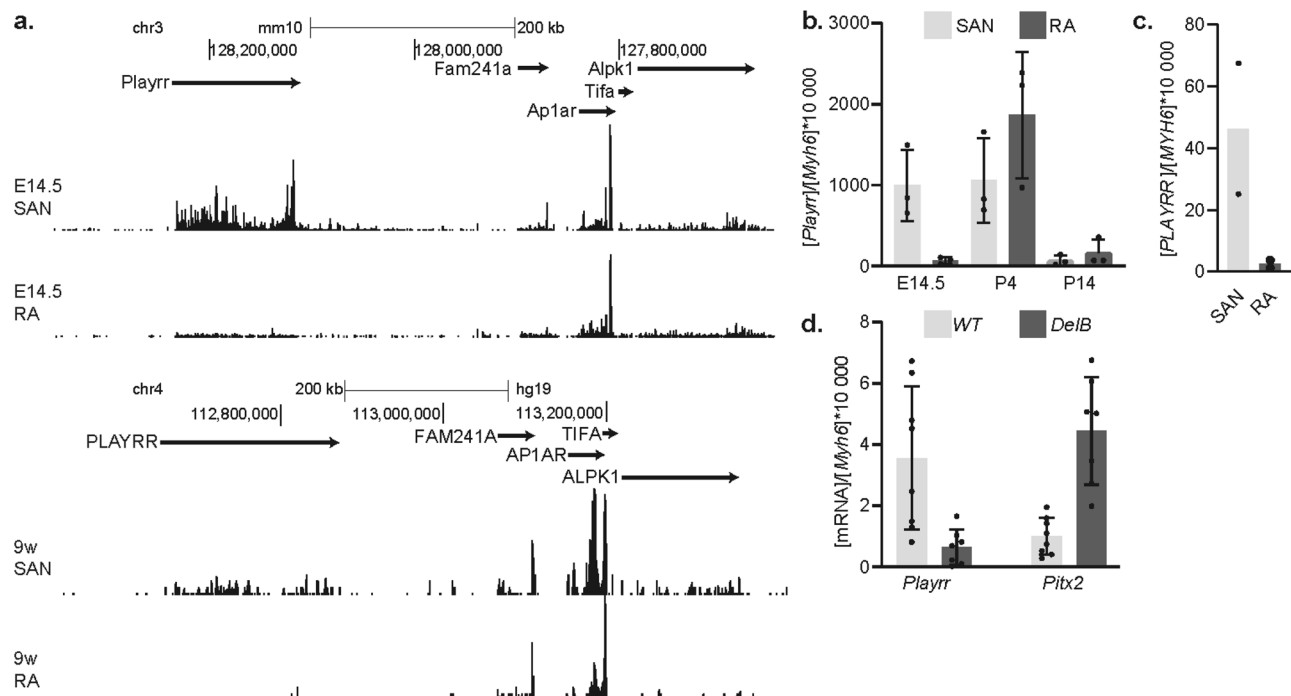

**Fig. 5 | The mutually exclusive expression of the lncRNA *Playrr* and *Pitx2* is disrupted in the *DelB* SAN. a** RNA-seq datasets showing that *Playrr* is expressed in the embryonic day (E)14.5 sinoatrial node (SAN) and less so in the E14.5 right atrium (RA) and in the prenatal human SAN at 9 weeks. Genome tracks were generated using UCSC Genome browser with the reference genome assembly *mm10*. **b** *Playrr* expression level normalized to *Myh6* in the mouse SAN (*n* = 3) and RA (*n* = 3) at

E14.5, postnatal day P4 and P14 (data from ref. 20). Error bars represent SD. **c** *PLAYRR* expression level normalized to *MYH6* in the human SAN (*n* = 2) and RA (*n* = 2) at 9/12 weeks of gestation (data from ref. 28). **d** *Playrr* and *Pitx2* expression in adult WT (*n* = 8) and *DelB* (*n* = 7) micro-dissected SAN normalized to *Myh6*. Error bars represent SD. Source data are provided as a Source Data file.

expression as a consequence of the deletion of CTCF binding sites and 3D chromatin remodeling. Of note, the deletion of CTCF binding sites was not found in 39 French patients presenting lone sinus node defects (*n* = 11), or with other electrical cardiac defect (*n* = 28) nor in 10,994 atrial fibrillation cases from the TOPMed-CCDG dataset with structural variant data[35] (Supplementary Data 10). Major progress in clinical management of these patients can be expected with an earlier diagnosis thanks to the molecular screening that can be considered as a major criterion for the diagnosis. Symptoms of this new cardiac entity at risk of sudden cardiac death can indeed be identified at an early stage of life such as in utero sinus node dysfunction. Our study also demonstrates the relevance of exploring the full spectrum of variants in coding and non-coding regions. This is illustrated by Family#6 in which we identified a rare *ANK2* chr4:114269433-A-G; *Hg19*; p.E1425G coding variant in 2003 by exome sequencing that co-segregates with the CTCF binding sites deletion[14]. As recently suggested, despite the ANK2 loss of function variants, the lack of penetrance observed in familial cases associated with the absence of family history in many cases may imply the involvement of additional genetic and/or environmental factors[27].

Of note, exonic coding mutations in *PITX2* and potentially upstream regions can lead to Axenfeld-Rieger syndrome characterized by anterior segment dysgenesis, retina and dental features[36]. Interestingly, the phenotype limited to cardiac tissue observed in this new clinical entity points to new cardiac tissue-specific regulatory mechanisms mediating tissue-specific dysregulation of *PITX2* expression in cardiac compartments and compartment-specific PITX2 target gene expression remodeling underlying the complex phenotype observed in deletion carriers.

We show that the deletion of the conserved boundary region between the *Pitx2*-containing TAD and adjacent ncRNA-containing TAD specifically disrupts the spatial regulation of *Pitx2* expression in *DelB* mice, resulting in ectopic expression of *Pitx2* in the SAN region and

reduced expression in the ventricles. Cardiac expression of other coding genes in the genomic vicinity of the deletion was largely unaffected, which is consistent with the notion that *Pitx2* is the only coding gene within the fused TADs. *Pitx2* expression is strictly spatiotemporally regulated during embryonic development and plays a key role in left-right asymmetry morphogenesis of several organs including the heart, stomach and gut[9]. In the developing heart, *Pitx2* is expressed selectively in the left-sided (derived) parts including the left atrium, left sinus venosus and pulmonary vein[37,38]. Mouse embryos deficient in *Pitx2* develop right isomerism of the atria, form a left-sided SAN, and fail to suppress aspects of the SAN-specific genetic program at the left side[38-40]. Conversely, here we see the severe disruption of the SAN-specific genetic program in the right-sided *DelB* SAN region coinciding with the induction of ectopic *Pitx2* expression in the SAN. Among the suppressed genes in *DelB* SAN tissue are genes encoding the key pacemaker transcription factors Tbx3 and Isl1 and ion channels that drive pacemaker function including Hcn4. Both Tbx3 and Isl1 activate and maintain the pacemaker genetic program in the heart, preventing the expression of genes associated with the working atrial myocardium in the SAN domain and their loss compromises SAN development, culminating in SAN dysfunction[20,21,25,26]. Cardiac-specific ablation of Hcn4 expression in mice results in severe and lethal bradycardia[41,42]. We confirmed the loss of the pacemaker genetic program in the *DelB* SAN by comparing differentially expressed genes with those enriched in pacemaker cardiomyocytes[22]. We found additional gain of atrium-specific markers in the *DelB* SAN tissue. The loss of SAN-associated gene expression, as indicated by a marked reduction in *Isl1* and *Tbx3* expression, in adult *DelB* mice indicates that *Pitx2* is not only necessary, but also sufficient to suppress the SAN genetic program.

Tbx5 and Pitx2 form an incoherent feed-forward loop that is driven by Tbx5 and suppressed by Pitx2, governing a left atrial transcriptional network that regulates rhythm effector gene expression[43].

Reduced Tbx5 expression in adult mice disrupts the expression of calcium handling and AF-susceptibility genes, reducing myocardial automaticity and elevating atrial fibrillation susceptibility, these phenotypes can be rescued by Pitx2 haploinsufficiency. While Tbx5 expression seems unaffected in the *DelB* SAN, the Tbx5-Pitx2 balance may be disrupted in the SAN due to the ectopic expression of *Pitx2*. We see that some, but not all, previously identified targets of the Tbx5-Pitx2 regulatory loop, including those associated with calcium handling (*Ryr2*, *Pln*) and rapid depolarization (*Scn5a*) are differentially expressed in the *DelB* SAN, potentially driving the susceptibility to atrial arrhythmias and SAN dysfunction in *DelB* mice.

The *PITX2* locus has been associated with atrial fibrillation[12], the most significant atrial fibrillation susceptibility locus lying 170 kb upstream of *PITX2* in the noncoding gene desert on chromosome 4q25[44]. This region houses conserved enhancers that can interact with the *Pitx2* promoter and predispose mice to atrial fibrillation upon deletion[45,46]. Here, we further illustrate the role of the gene desert on chromosome 4q25 in the finely tuned spatial regulation of *PITX2* expression in the heart, impacting on atrial fibrillation susceptibility and SAN dysfunction.

Non-coding regions enriched for epigenetic signatures associated with gene regulation predict the location of *cis*-regulatory elements (RE) that drive tissue- and context-specific gene expression[47]. While REs and their target genes are often confined to the same TAD, their tissue-specific functional range can span entire gene deserts[2,48,49]. For example, ablation of a regulatory element 1.1 Mb distal from *Tbx3* abolishes *Tbx3* expression in the SAN while its expression in other tissues is preserved[28]. We have interrogated the regulatory landscape of the chr4q25 locus in ventricular, atrial and hiPSC-derived pacemaker-like cardiomyocytes and found a number of candidate pacemaker-specific REs that may mediate the ectopic activation of *PITX2* in the SAN region. Furthermore, we observed that the mutually exclusive expression of the lncRNA *Playrr*, located in the noncoding TAD, and *Pitx2*, previously described in the developing dorsal mesentery and implicated in the left-right asymmetry of the gut[18], is mirrored in the pre- and postnatal mouse heart. Moreover, we identified expression of a human lncRNA, which based on conservation of location, promoter sequence and expression pattern is likely the human orthologue of *Playrr*, in the fetal human SAN. We noted a marked reduction in *Playrr* expression in the micro-dissected SAN of adult *DelB* mice, complimenting the SAN-specific gain in *Pitx2* expression. This observation indicates the *Pitx2-Playrr* mutual repression mechanism described in gut development[18] may also be involved in *Pitx2* suppression in the SAN. We suggest that the loss of a boundary between the *PITX2*-TAD and the noncoding RNA-TAD permits ectopic interactions between the initially spatially separated *Pitx2* promoter and SAN-specific regulatory elements and/or *Playrr*, leading to ectopic activation of *Pitx2* expression in the SAN.

In summary, we identified rare deletions affecting a TAD boundary that induce chromatin conformation remodeling. The native interactions between *PITX2* and its regulatory regions are modified in favor of new interactions with regulatory elements present in the TAD adjacent to the *PITX2*-TAD. *PITX2* deregulation is characterized by an opposite response between cardiac compartments: increased *PITX2* expression in the SAN region (ectopic activation) and reduced expression of *PITX2* in the ventricle. This singular molecular mechanism underlies the complex cardiac phenotype observed in patients including SAN dysfunction, atrial fibrillation and developmental defects. Exploring TAD boundary impairment in unresolved inherited disorders may reduce diagnostic wandering and enhance our understanding of the molecular mechanisms underlying human pathophysiology.

## Methods

### Patients recruitment and study design
The study was conducted according to international recommendations. Written consent was systematically obtained before clinical and genetic analysis was performed in these patients. This research project has been approved by the French Minister of education and research as well as by the personal protection committee (DC-2011-1399). Accordingly no compensation has been given in return of their participation to the study. Families clinical, electrocardiographic, echocardiographic, and genetic data were collected prospectively and analyzed retrospectively.

Inclusion criteria were families presenting forms of sinus dysfunction defined as the presence of sinus bradycardia (daytime heart rate <50 BPM), diurnal sinus pause >3 s, or chronotropic incompetence objectified during an exercise test; cardiac morphological abnormalities such as atrial septal defect, left ventricular non-compaction, mitral valve prolapse or billowing, or an abnormal systemic venous return and other cardiac rhythm disorders and in particular atrial fibrillation or ECG abnormalities (morphological abnormality of the T wave or prolongation of the QT duration). No disbalance has been noticed in the recruitment between Female (53%) and Male (47%) patients ($p = 0.5102$, Chi-squared test).

Quantitative variables were presented by their mean and standard deviation or by their median and minimum and maximum values according to their distribution. Categorical variables were presented in numbers and percentages. The statistical software used was SAS version 9.4. The threshold of significance was set at 0.05.

### Whole genome sequencing and analysis
From probands of Families#1,4,5,6 and 7, DNA was extracted from peripheral blood. In total, 1.1 µg has been used to prepare a library for whole genome sequencing (WGS), using the Illumina TruSeq DNA PCR-Free Library Preparation Kit, according to the manufacturer's instructions. After normalization and quality control, qualified libraries have been sequenced on an Illumina HiSeqX5 system (Illumina Inc., CA, USA), as paired-end 150 bp reads. Each sample reaches a minimum sequencing depth of 80% and coverage at ≥20x.

The WGS pipeline followed the recommendations of the Broad Institute GATK "Best Practices". In brief, reads were mapped to the GRCh37 human genome using bwa-mem, duplicates were marked using picard and GATK was used to realign and recalibrate the reads (broadinstitute.org/gatk/guide/bestpractices.php). Structural variations were called using the LUMPY framework[50]. Structural variants were detected using lumpy-sv express v0.2.13. The profiles of depth were plotted using lowresbam2raster from the jvarkit package (https://doi.org/10.6084/m9.figshare.1425030).

For Japanese family#2 WGS has been outsourced of Macrogen Japan Corp, and they analyzed by using HiSeq sequencer (illumine corp.)

For Japanese family#3 the deletion was determined using multiple qPCRs to estimate the deletion size on chr 4q25. Classic qPCR was performed on genomic DNA, using syber green (Thermo Fisher Scientific) and primers at 0.2 µM listed in Supplementary Data 11 with standard cycling parameters: 94 °C for 2 min, 40 cycles at 94 °C during 15 s and 60 °C during 1 min.

The deletion breakpoints of all individuals (depending on available DNA) were validated with classic PCR protocol using specific primers for each family (Supplementary Data 12) followed by Sanger sequencing. Except for Family#7 which shows repeated region at the 5' end preventing the determination of the breaking point by sanger sequencing. An approximate deletion size was obtained by WGS (Supplementary Fig. 1).

### qPCR and WGS to interrogate the prevalence of the deletion in patients presenting lone electrical cardiac defect
qPCR was performed on genomic DNA of patients recruited at Nantes, France, presenting lone SND, Long QT syndrome type 4 like. qPCR was performed using syber green (Thermo Fisher Scientific) and a set of primers at 0.2 µM: 4q25_112563000 listed in Supplementary Data 11 with standard cycling parameters: 94 °C for 2 min, 40 cycles at 94 °C during 15 s and 60 °C during 1 min.

The prevalence of structural variants (SV) overlapping with the CTCF binding sites (contained in the minimal deletion) among atrial fibrillation cases and healthy controls was assessed by interrogating the largest whole-genome sequence (WGS) dataset for atrial fibrillation, which was produced through the NHLBI Trans-Omics for Precision Medicine (TOPMed) and the Centers for Common Disease Genetics (CCDG) consortia as previously described[35]. This SV call-set contained data on 10,994 atrial fibrillation cases and 25,505 controls (after SV quality-control procedures), which were assembled from 28 individual cohorts with various sampling designs (including prospective population-based cohorts, early-onset atrial fibrillation cohorts, lone atrial fibrillation cohorts and all-comers atrial fibrillation case-control cohorts[35]).

In this call-set, SVs (from short-read WGS) were identified in each sample separately using the Parliament2 pipeline[51], which provides a union of calls from six different programs (BreakDancer, BreakSeq, CNVnator, Delly, Lumpy and Manta). The site lists were merged across all samples using survivor[52,53] and filtered using SVTyper[53]. Final genotypes were then called in each sample using all sites from the merged variant list with the muCNV software (Goo Jun, unpublished). Association testing for SVs was performed using a logistic mixed-effects model implemented in custom software[54] (https://github.com/seanjosephjurgens/UKBB_200KWES_CVD; branch v1.2).

### Ethics statement
Housing, husbandry, all animal care, and experimental protocols were in accordance with guidelines from the Directive 2010/63/EU of the European Parliament and Dutch government. Protocols were approved by the Animal Experimental Committee of the Amsterdam University Medical Centers. Animal group sizes were determined based on previous experience.

### Generation of mutant mice
Mutant mice were generated using CRISPR/Cas9. Guide RNA (sgRNA) target sites were designed with ZiFiT Targeter (Supplementary Data 13). The Alt-R® CRISPR-Cas9 crRNA, Alt-R® CRISPR-Cas9 tracrRNA (IDT #1072532) and Alt-R® S.p. Cas9 Nuclease V3 (IDT #1081058) were ordered at IDT. One-cell FVB/NRj zygotes were microinjected via the "Mouse zygote microinjection Alt-R™ CRISPR-Cas9 System ribonucleoprotein delivery protocol" with 20 ng/μl Cas9 Nuclease and 10 ng/μl guide RNA to generate mouse founders. Deletions were validated by PCR (Supplementary Data 14) and Sanger sequencing. Breakpoints for the *DelA*, *DelB* and *DelC* mouse models were: chr3:128,346,349-128,357,670, chr3:128,329,001-128,346,349 and chr3:128,270,995-128,328,982, respectively. *DelA* mouse line was generated using guides 3 and 4, B deletion line with guides 2 and 3, and the C deletion line with guides 1 and 2. To obtain stable lines, the founders were backcrossed with WT FVB/NJ mice and maintained on a FVB/NJ background ordered from the Jackson laboratory (#100800). For tissue harvest, animals were euthanized by 20% $CO_2$ inhalation followed by cervical dislocation.

### In vivo electrophysiology
Twelve to 20-week-old male and female mice were anesthetized with 5% Isoflurane (Pharmachemie B.V. 061756) and placed on thermostated mat (36 °C) with a steady flow of 1.5% isoflurane during all experiments. Electrodes were inserted subcutaneously in the limbs and connected to an ECG amplifier (Powerlab 26T, AD Instruments). The electrocardiogram (ECG) was measured for 5 min. ECG parameters were determined in Lead II (L-R) based on the last 60 s of the recording. For atrial stimulation, an octapolar CIB'er electrode (NuMED) was advanced through the esophagus to achieve atrial capture. Atrial capture thresholds were determined for each mouse, and all pacing protocols were performed at 2x threshold. For sinus node recovery time (SNRT) measurements, a 4 s pacing train with a cycle length of 120 or 100 ms was used. SNRT was defined as the interval between the last

pacing stimulus and onset of the first P wave. To control for differences in sinus rate, SNRT was normalized to resting heart rate (cSNRT = SNRT − RR interval). To determine the Wenckebach cycle length (WBCL) we applied a 4-s pacing train starting at a cycle length of 100 ms, and decreasing by 2 ms until atrioventricular (AV) block was first observed. Atrial arrhythmia (AA) was induced by 1 or 2 s bursts starting with a cycle length of 60 ms, decreasing successively with a 2-ms decrement, down to a cycle length of 10 ms. AA duration was the sum of time each mouse spent under and AA episode after completion of the two passes. Atrial arrhythmia (AA) inducibility was scored as the number of mice in which at least one episode lasting >1 s of arrhythmia was induced after pacing. Mice of both sexes were used for atrial arrhythmia induction experiments.

### Histology
Hematoxylin and eosin (H&E) staining was performed on paraffin sections using standard methodology and staining reagents.

### Atrial septation assay
Ten to 20-week-old mouse hearts were dissected in PBS warmed to 37 degrees. Ventricles and the lobe of the left atrium were removed. Incomplete atrial septation was assigned if there was passage of Orange G solution from one atrium to the other after pressurization of the intact atrium, achieved by pulsing dye using a 28G needle.

### qPCR mouse model
Total RNA was isolated from atria, ventricle and kidney from P21 male and female mice using ReliaPrep RNA Tissue Miniprep System (Promega, Z6112) according to the manufacturer's protocol. cDNA was reverse transcribed with oligo dT primers from 500 ng of total RNA, or random hexamers from 500 pg of CM nuclear RNA, according to the manufacturer's protocol of the Superscript II Reverse Transcriptase system (Thermo Fisher Scientific, 18064014). Expression levels of candidate target genes were determined by quantitative real-time PCR using a LightCycler 480 Instrument II (Roche Life Science, 05015243001). Expression levels were measured using LightCycler 480 SYBR Green I Master (Roche, 04887352001) and the primers had a concentration of 1 pmol/l (Supplementary Data 15). The amplification protocol consisted of 5 min 95 °C followed by 45 cycles of 10 s 95 °C, 20 s 60 °C and 20 s 72 °C. Relative start concentration (N0) was calculated using LinRegPCR[55]. Values were normalized to the geometric mean of two reference genes per experiment (Hprt, Ppia, or Rpl32)[56].

### iPSC generation and cardiomyocyte ventricular like differentiation
The human-induced pluripotent stem cells (hiPSC) were derived from the Peripheral Blood Mononuclear Cell reprogrammed using Sendai virus in the iPSC core facility of Nantes University. hiPS cells were obtained from a man aged 55 years previously described[57] who had no clinical symptoms and normal ECG (TXi006-A). hiPSC were maintained on stem cell-qualified Matrigel-coated plates (0.1 mg/ml BD Bioscience) and cultured in StemMACS iPS-brew XF medium (Miltenyi Biotec). hiPSC were passed every 3–4 days, using Gentle Cell Dissociation reagent (StemCell Technologies).

To generate ventricular like CMs, hiPS cells were cultured as a monolayer in StemMACS iPS-brew (StemMACS) supplemented with Y-27632 ROCK inhibitor 1/1000 (Stemcell Technologies). When hiPS cells reached 70–80% confluence, they were differentiated into hiPSC-CMs using the previously described "Matrix sandwich" Method[58]. At day 10 of differentiation, beating hiPSC-CMs were purified following the glucose starvation protocol[59] using RPMI1640 medium no glucose (Life Technologies). By day 17, hiPSC-CM were cultured in RPMI1640 medium supplemented with B27 complete (Life Technologies), 1X L-glutamine and 1% NEAA and changed every 2 days until the end of the differentiation day 30.

## iPSC generation and cardiomyocyte sinoatrial like differentiation

As a control female hiPSC line, 201B7 were used in present study[60]. These hiPSCs were cultured as described[61].

To generate sinoatrial nodal-like CMs (SAN-CMs), we followed an embryoid body (EB) approach by using culture media and appropriately adding BMP4, ROCK inhibitor Y-27632, Activin A, bFGF, the WNT inhibitor IWP2 or IWP3, the Activin/Nodal/TGFβ signaling inhibitor SB-431542, Retinoic acid, VFGF and the FGF signaling inhibitor PD 173074[62]. These EBs were cultured in 96-well ultra-low attachment dishes in which EBs were aggregated at day 0, and they were dissociated into single cells and reaggregated at day 3.

## iPSC genome editing−isogenic model

Fifteen kb deletion and the open region containing the CTCF binding domain deletion from the control hiPS cell line were performed using the Alt-R™ CRISPR-Cas-9 system previously described[63]. Briefly, guides RNA (gRNA; Integrated DNA Technologies (IDT)) were designed to target the extremes of the deletion of the family 1 and the open region containing the CTCF binding sites (Supplementary Data 16). hiPSC were transfected with the Alt-R CRISPR-Cas9 complex using Amaxa P3 Primary Cell 4D-Nucleofector® X Kit (Lonza, #V4XP-2024). ATTO488-positive cells were sorted into 96-well plates to obtain clonal populations using a FACSMelody™ (BD Biosciences). This first genome edition generated hiPSC 15 kb +/− clones and hiPSC CTCF + /− clones. A second genome edition on these heterozygous clones was then carried out in the same way to obtain hiPSC 15kb−/− and hiPSC CTCF−/− clones. For genotyping isogenic models, hiPSC were lysed using 30 µl QuickExtract solution (Epicentre, #QE09050) and DNA was extracted according to the manufacturer's instructions. PCR were performed to genotype the 15 kb region and the open region containing the CTCF (Supplementary Data 17) followed by Sanger sequencing (Eurofins genomics)

To ablate a 19-bps CTCF binding domain (CTCF-BD) on both alleles of control hiPSC line: 201B7, by using CRIPR-Cas9 system6, we designed two different synthetic gRNA (IDT) on IDT website ( https://sg.idtdna.com/site/order/designtool/index/CRISPR_CUSTOM ) (Supplementary Data 16). hiPSCs dissociated to single cells by Accumax (Innovative Cell Technologies, San Diego, CA, USA) were transfected with two gRNAs and RNP complex mixed with Alt-R Cas9 Nuclease 3NLS (IDT) and crRNA:tracrRNA duplex (IDT), by using the NEPA 21 electroporator (NEPA GENE). After transfected colonies were picked up, genomic DNA was isolated by using the Geno Plus Genomic DNA Extraction 35 Miniprep System Kit (Viogene Bio Tek Corp). Purified DNA was amplified with specific primers (Supplementary Data 17) and analyzed by Genetic Analyzer 3130 and Big Dye Terminator v3.1 (Thermo Fisher Scientific) and then, sanger sequencing was performed to confirm the targeted site and possible off-target sites.

## ATAC-seq

ATAC-seq experiments were performed in duplicated on the hiPSC-CM WT. hiPSC-CM were dissociated at day 28–30 in single cells suspension using accutase solution (Sigma, #A6964). In total, 50,000 cells were washed by a centrifugation in cold PBS. Pelleted cells were resuspended in the transposase reaction mix supplemented with digitonin (25 µl 2x TD buffer, 5 µl TDE1 (Illumina), 1 µl digitonin 0.5% (Promega) and 19 µl of nuclease free water). The transposition reaction was carried out for 30 min at 37 °C. The samples were immediately purified using the DNA Clean and Concentrator-5 kit (Zymo Research).

Following purification, each sample was dual indexed with i7 and i5 primers from the Nextera Index Kit (Illumina) and amplified using Nextera DNA library Prep Kit (Illumina) using the following PCR conditions: 72 °C for 3 min, 98 °C for 30 s, followed by 11 cycles at 98 °C for 10 s, 63 °C for 30 s and 72 °C for 3 min. Libraries were double size selected with Beckman Coulter™ Agencourt AMPure XP (0.5X and 1.3X ratios). Library quality was assessed using TapeStation and libraries

were quantified using NEBNext® Library Quant Kit for Illumina® (New England Biolabs). Sequencing was performed on an Illumina Nova-Seq6000 platform in 2 × 51 paired-end mode. Trimming and demultiplexing was done using Illumina bcl2fastq2.

Read mapping was done using bowtie2[64] with paired end settings and "-X 2000" on *hg19*. Only properly mapped reads were kept, duplicated reads were filtered using samblaster[65]. Bams were converted into bedpe format and coordinate shifted using deeptools[66] (--ATACshift option). Broad peak calling was carried out with MACS2 using bedpe options and default *p* value and *q* value for broad cutoff. Quality control was done by computing the fraction of reads in peaks (FrIP) with subread feature counts using peaks called with MACS2 narrowPeak. Complexity of the library was evaluated according to standards defined by the ENCODE consortium[67] with the NRF (nonredundant fraction), PCB1 and PCB2 (PCR bottleneck 1 and 2) computed for all samples. TSS enrichment was also checked using ENCODE methods as defined in the ENCODE atac-seq pipeline[68]. Correlation between samples was evaluated with deeptools plotCorrelation on all filtered reads of chromosome 1.

## CUT&RUN assays

For each reaction, 100,000 hiPSC-CMs were dissociated at day 28–30 in single cells suspension using accutase solution (Sigma, #A6964) for 8 min in 37 °C. Cells were resuspended in 20% fetal bovine serum/RPMI1640 medium supplemented with B27 complete (Life Technologies) and centrifugated. CUT&RUN was performed following the manufacturer's protocol (Cell Signaling technology##39915). The positive control antibody mAb against trimethylated histone H3K4 (H3K4me3, 1:50, CST#9751T) and the negative control antibody mAb IgG isotype (1:50, CST#66362) was provided in the kit. Antibodies mAb against acetyl H3K27 (H3K27ac,1:100, CST#39133), Monomethylated H3K4 (H3K4me1, 1:50, CST#39297) and CTCF (1:50, CST#61311) were also used. DNA was purified using a DNA purification kit (New England Biolabs#T1030).

CUT&RUN libraries were prepared using the NEBNext® Ultra™ II DNA Library Prep Kit for Illumina® (E7645, NEB) according to the manufacturer's instructions except for the End Prep step which took place for 30 min at 20 °C followed by 1 h at 50 °C in order not to denature the small fragments. The adapter-ligated DNA was cleaned up with Beckman Coulter™ Agencourt AMPure XP without size selection. Following purification, each sample was dual indexed with NEBNext Multiplex Oligos for Illumina and amplified using the following PCR conditions: 98 °C for 30 s, followed by 12 cycles at 98 °C for 10 s, 65 °C for 15 s and a final extension at 65 °C for 5 min. Library quality was assessed using TapeStation and libraries were quantified using NEB-Next® Library Quant Kit for Illumina® (New England Biolabs). Sequencing was performed on an Illumina NovaSeq6000 platform in 2 × 50 paired-end mode.

Read mapping was done using bowtie2[64] with paired end settings and "-I 0 -X 2000 --local --dovetail" on *hg19*. Only properly mapped reads were kept, duplicated reads were filtered using samblaster[65]. Complexity of the library was evaluated according to standards defined by the ENCODE consortium[67] with the NRF (nonredundant fraction), PCB1 and PCB2 (PCR bottleneck 1 and 2) computed for all samples. TSS enrichment was also checked using ENCODE methods as defined in the ENCODE atac-seq pipeline[68]. Correlation between samples was evaluated with deeptools plotCorrelation on all filtered reads of chromosome 1. Normalization was performed using the ratio of spike-In DNA. Spike-In DNA was quantified by aligning samples onto *sacCer3* using bowtie2 with parameters "--end-to-end --very-sensitive --no-mixed --no-discordant --no-overlap --no-dovetail". Quality control was done by computing the fraction of reads in peaks (FrIP) with subread featureCounts[69] using peaks called with SEACR[70]. Motif enrichment analysis was performed using MEME-Chip[71], using all common peaks between conditions as a background.

## in situ Hi-C

Hi-C libraries were processed using our in-house in situ version published elsewhere[72]. 1x10E6 hiPSC-CMs (per sample hiPSC-CM WT and hiPSC-CM 15Kb−/−) were fixed in 2% formaldehyde, lysed, and digested overnight with the four basepair cutter DpnII (New England BioLabs, #M0202). PCR-amplified libraries were sequenced in total for 320 million fragments (two replicates per sample) in a 100 bp paired-end run on a NovaSeq6000 (Illumina).

Hi-C Paired-end sequencing data was processed using the pipeline described in Melo et al.[73] using the following bioinformatic tools: Juicer for processing Hi-C reads[73]; BWA v0.7.17[74] for aligning the Illumina reads to reference genome *hg19*; Juicebox for Hi-C maps visualization[75]; and Knight and Ruiz (KR) for Hi-C normalization[76]. For the generation of Hi-C maps, we used read-pairs with mapping quality (MAPQ) ≥ 30.

## RNA-sequencing

hiPSC-CM WT and hiPSC-CM CTCF−/− total RNA of 8 differentiations per cell line were isolated using the NucloSpinRNA kit (Macherey-Nagel #740955.50) following the manufacturer's instructions. Quality and integrity of RNA samples were assessed using the 2100 Bioanalyzer and RNA 6000 Nano LabChip kit series II (5067-1511, Agilent Technologies). We used the Illumina® Stranded mRNA Prep, Ligation kit (20040532, Illumina) to purify and capture the mRNA molecules containing polyA tails with oligod(T). The purified mRNA is fragmented and copied into first strand complementary DNA (cDNA) using reverse transcriptase and random primers. In a second strand cDNA synthesis step, dUTP replaces dTTP to achieve strand specificity. The final steps add adenine (A) and thymine (T) bases to fragment ends and ligate adapters. The resulting products are purified and selectively amplified for sequencing on an Illumina system. Fragment size of libraries was controlled on D1000 ScreenTape with 2200 TapeStation system (Agilent Technologies). Dual-indexed libraries were quantified by Qubit Quantification using Qubit 1X ds DNA HS Assay kit (Q33231, Life Technologies). Each library was pooled and prepared according to Illumina® Stranded mRNA Prep, Ligation kit protocol (Document # 1000000124518 v00, Illumina). Paired-end sequencing (2 × 101 cycles) was carried out with a run on NovaSeq 6000 Sequencing System

Bioinformatics steps are performed using a snakemake[77] pipeline (https://bio.tools/DEPIB). Raw paired-end fastq files are processed with cutadapt (1.18) and prinseq (0.20.4) in order to remove Illumina adapters and low-quality reads. Alignment on genome reference, available from the ensembl ftp download website, is performed using STAR (2.7.3a). htseq-count is used to count the number of reads for each gene in each sample. The expression matrix is normalized and differentially expressed genes (DEG) are searched using the R package deseq2 ("1.30.1" R version:4.0.2)[78]. If DEGs are found, functional annotation is performed using the R package ClusterProfiler[79,80].

Total RNA isolation from microdissected adult SAN tissue samples was performed using the ReliaPrep RNA Tissue Miniprep System (Promega, Z6112) according to the manufacturer's instructions. In total, 100−200 ng of SAN RNA was used for library generation using the KAPA mRNA HyperPrep kit (Roche) and sequenced on the HiSeq4000 system (Illumina) with 50 bp single-end reads. RNAseq sample sizes are as follows: 7 *DelB* SAN and 8 WT SAN. Reads were mapped to *mm10* build of the mouse transcriptome using STAR. The DESeq2 package based on a negative binomial distribution model was used to perform differential expression analysis. *p* Values were corrected for multiple testing using a false discovery rate (FDR) method of Benjamini−Hochberg using a FDR of 0.05 as FDR control level.

## RTqPCR isogenic model

Total RNA was isolated using the TRIzol Reagent (Thermo Fisher Scientific, Waltham, MA, USA) from EBs at day 20. These RNAs were reverse-transcribed into complementally DNA (cDNA), using Verso cDNA synthesis Kit (Thermo Fisher Scientific). The real-time qPCR was performed using Taqman method with PrimeTime Gene Expression Master Mix with appropriate primers and probes for each gene (Integrated DNA Technologies, IL, USA). The PCR-related primers list is provided in Supplementary Data 18. The expression of interest genes was normalized to that of ACTB and each gene expression level was determined according to the ΔΔCt method

## Statistical analysis

For the mouse model, the experimenters were blind to mouse genotype during all measurements and outcome assessment. Data sets were tested for normality using the Shapiro−Wilk test. Whole tissue real-time quantitative polymerase chain reaction fold changes from WT were analyzed across four genotypes (WT, *DelA, DelB, DelC*) and within each tissue type using Kruskal−Wallis test followed by pairwise comparisons with Dunn's multiple comparison tests in right and left atria, and Welch's ANOVA followed by Dunnett's T3 multiple comparison tests in ventricles of line B.

In vivo electrophysiology was analyzed with Welch's *t* test for RR and PR intervals, heart rate-corrected sinus node recovery time (cSNRT), Wenckebach cycle length (WBCL); Mann−Whitney tests for heart rate variation (SDNN), QRS interval, and atrial arrhythmia (AA) duration; Fisher's exact test was used for AA inducibility].

In vivo electrophysiology was analyzed with Welch's ANOVA followed by Dunnett's T3 multiple comparison test for RR and sex-corrected PR intervals, and heart rate-corrected sinus node recovery time (cSNRT); Kruskal−Wallis test followed by Dunn's multiple comparison test for heart rate variation (SDNN), QRS interval, and atrial arrhythmia (AA) duration; ANOVA followed by Tukey's multiple comparison test for Wenckebach cycle length (WBCL); and Chi square for AA inducibility.

Heart-to-body weight ratio differences were determined with unpaired *t*-tests and atrial defects were analyzed with Fisher's exact test. Multiple testing corrections were performed independently within each hypothesis. Data are presented as individual data points and mean and *p* < 0.05 defines statistical significance. Statistical analysis was performed using GraphPad Prism 9.

For the hiPSC model, all sample numbers are represented by cell culture replicates. For statistical comparisons, Student's *t* test, paired *t*-tests or ANOVA were appropriately employed and followed by Tukey's test if an ANOVA found differences among groups. A *p* value <0.05 was considered statistically significant.

## Reporting summary

Further information on research design is available in the Nature Portfolio Reporting Summary linked to this article.

# Data availability

The NHLBI Trans-Omics for Precision Medicine (TOPMed) are available at https://topmed.nhlbi.nih.gov, the Centers for Common Disease Genetics (CCDG) at www.genome.gov/Funded-Programs-Projects/NHGRI-Genome-Sequencing-Program/Centers-for-Common-Disease-Genomics and CARE at http://ns104190.ip-147-135-44.us/CARE_portal. Individual-level data (WGS) sharing is subject to restrictions imposed by patient consent and local ethics review boards. ATAC-seq data from the CARE database are available at http://ns104190.ip-147-135-44.us/CARE_portal. RNA-seq datasets (SAN and RA regions) obtained from mouse model are accessible on GEO: GSE242698 and single cell RNA-sequencing datasets are available on GEO: GSE132658. Fetal mouse SAN and RA RNA-sequencing datasets are available on GEO: GSE65658, fetal human SAN and RA RNA-sequencing datasets are available at GSE125932. ATAC-seq and CUT&RUN datasets obtained from cardiomyocytes derived iPSc are available on GEO: GSE243902. Other datasets generated during and/or analyzed during the current study can be

made available upon reasonable request to the corresponding authors. Source data are provided with this paper.

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

## Acknowledgements

The authors are greatly indebted to the patients included in the study. They thank Valérie Cotard, Marie-France Le Cunff, Noémie Bourgeais, Marie Marrec, Guenola Coste, Sakai Yoshiyuki and Takuro Misaki for assistance in patient recruitment and follow-up and Aurore Girardeau, Amandine Caillaud, Leander Beekman, Takeshi Hatani, and Kyoko Yoshida for their technical support. We thank the biological resource centre for biobanking (CHU Nantes, Nantes Université, Centre de ressources biologiques (BB-0033-00040), F-44000 Nantes, France) applying following guidelines[81]. We are most grateful to the Genomics Core Facility GenoA, member of Biogenouest and France Genomique and to the Bioinformatics Core Facility BiRD, member of Biogenouest and Institut Français de Bioinformatique (IFB) (ANR-11-INBS-0013) for the use of their resources and their technical support. We are grateful to the support of the GenOmic variability in heaLth & Disease "GOLD", a national transversal research project supported by INSERM. J.B. is supported by the research program Etoiles montantes des Pays de la Loire REGIOCARD RPH081-U1087-REG-PDL, ANR JCJC LEARN (R21006NN, RPV21014NNA) and the Programme "*High throughput sequencing and rare diseases*" of the "Fondation maladies rares". M.B. was supported by IRP- VERACITIES—New Mechanisms for VEntricular ARrhythmia and CardIomeTabolic DIseasES, an I-SITE NExT health and engineering initiative (Ecole Centrale & Nantes University), by the IRP—GAINES—Genetic Architecture IN cardiovascular disEaSes funded by INSERM and CNRS and "Fondation pour la Recherche Médicale" (FRM: FDT202204014986). T.L.T. was supported by Fédération Française de Cardiologie (2015, Paris, France), Fondation Coeur et Recherche (2015, Paris, France), French Ministry of Health "PHRC-I 2012" (API12/N/019, Paris, France). V.M.C. is supported by OCENW.GROOT.2019.029, Fondation Leducq (14CVD01) and CVON2019-002 OUTREACH. H.M., W.S. and T.I. was supported by JSPS KAKENHI *Grant* Number JP19K08566. H.M. was supported by Grants of Nippon Medical School and Japanese

Heart Rhythm Society with NIHON KOHDEN/St. Jude Medical Arrhythmia Fellowship program for Young Japanese Investigator. This work was supported by grants from: JSPS KAKENHI (JP19K08538 to T.Makiyama), a grant from Japan Agency for Medical Research and Development (AMED) (JP20bm0804022 to Y.Yoshida), the Suzuken Memorial Foundation (T.Makiyama and T.Kimura). This work was carried out within the context of the "SysMics Cluster" and benefited from state aid managed by the National Research Agency under the "France 2030" investment plan, financial support of the Pays de la Loire Region and Nantes Métropole. S.J.J. was also supported by an Amsterdam UMC Doctoral Fellowship and the Junior Clinical Scientist Fellowship (03-007-2022-0035) from the Dutch Heart Foundation. S.H.C. was supported by the BioData Ecosystem fellowship. P.T.E. supported by funding from the Fondation Leducq (14CVD01), by grants from the National Institutes of Health (1RO1HL092577, 1R01HL157635, 5R01HL139731) and by a grant from the American Heart Association (18SFRN34110082) and from the European Union (MAESTRIA 965286).

## Author contributions

M.B., H.M., F.M.B., U.S.M., T.A., P.L., L.V.D.M., T.M., S.M., V.M.C., V.P., J.-J.S. and J.B. conceived/designed elements of the study. All authors acquired, analyzed or interpreted data. M.B., H.M., F.M.B., U.S.M., T.A., P.L., L.V.D.M. F.P., T.M., S.M., V.M.C., V.P., S.L.S., R.R., N.M., T.L.T., J.-J.S. and J.B. drafted the manuscript. All authors critically revised the manuscript for important intellectual content and approved the final version.

## Competing interests

Y.Yoshida owns stock in iPS Portal. Takeda Pharmaceutical Company Ltd. is paying the salary of T.T. independently of this work. P.T.E. has received sponsored research support from Bayer AG and IBM Health, and he has consulted for Bayer AG, Novartis and MyoKardia. The remaining authors declare no competing interests.

## Additional information

[1]Nantes Université, CHU Nantes, CNRS, INSERM, l'institut du Thorax, F-44000 Nantes, France. [2]The Department of Cardiovascular Medicine, Nippon Medical School Hospital, Tokyo, Japan. [3]Department of Medical Biology, Amsterdam Cardiovascular Sciences, Amsterdam University Medical Centers, University of Amsterdam, 1105 AZ Amsterdam, The Netherlands. [4]Max Planck Institute for Molecular Genetics, RG Development and Disease, 13353 Berlin, Germany. [5]Department of Cardiovascular Medicine, Kyoto University Graduate School of Medicine, Kyoto, Japan. [6]Department of Medical Biology, Amsterdam Cardiovascular Sciences, Amsterdam Reproduction and Development, Amsterdam University Medical Centers, University of Amsterdam, 1105 AZ Amsterdam, The Netherlands. [7]Omics Research Center, National Cerebral and Cardiovascular Center, Suita, Japan. [8]Department of Cardiology, Fukuoka Children's Hospital, Fukuoka, Japan. [9]Cardiovascular Disease Initiative, Broad Institute of MIT and Harvard, Cambridge, MA, USA. [10]Department of Experimental Cardiology, Heart Center, Amsterdam Cardiovascular Sciences, Amsterdam UMC Location University of Amsterdam, Amsterdam, The Netherlands. [11]Department of Biostatistics, Boston University School of Public Health, Boston, MA, USA. [12]Department of Cell Growth and Differentiation, Center for iPS Cell Research and Application, Kyoto University, Kyoto, Japan. [13]Takeda-CiRA Joint Program for iPS Cell Applications, Fujisawa, Japan. [14]Department of Pancreatic Islet Cell Transplantation, National Center for Global Health and Medicine, Tokyo, Japan. [15]Department of Bioscience and Genetics, National Cerebral and Cardiovascular Center Research Institute, Suita, Japan. [16]Department of Cardiovascular Medicine, Shiga University of Medical Science, Ohtsu, Japan. [17]Cardiovascular Research Center, Massachusetts General Hospital, Harvard Medical School, Boston, MA, USA. [18]Demoulas Center for Cardiac Arrhythmias, Massachusetts General Hospital, Boston, MA, USA. [19]Service de Génétique Clinique, CHU Lille, Hôpital Jeanne de Flandre, F-59000 Lille, France. [20]University of Lille, EA 7364-RADEME, F-59000 Lille, France. [21]Unité de Cardiologie Pédiatrique, Hôpital des Enfants, F-31000 Toulouse, France. [22]Service de Cardiologie, GH La Rochelle, F-17019 La Rochelle, France. [23]Université Paris-Saclay, CEA, Centre National de Recherche en Génomique Humaine (CNRGH), 91057 Evry, France. [24]European Reference Network for Rare and Low Prevalence Complex Diseases of the Heart: ERN GUARD-Heart, Amsterdam, The Netherlands. [25]Department of Cardiology, Sapporo Teishinkai Hospital, Sapporo, Japan. [26]Cabinet Cardiologique, Clinique St. Joseph, F-16000 Angoulême, France. [27]Department of Community Medicine Supporting System, Kyoto University Graduate School of Medicine, Kyoto, Japan. [28]These authors contributed equally: Manon Baudic, Hiroshige Murata, Fernanda M. Bosada, Uirá Souto Melo, Takanori Aizawa, Pierre Lindenbaum, Lieve E. van der Maarel. [29]These authors jointly supervised this work: Takeru Makiyama, Stephan Mundlos, Vincent M. Christoffels, Vincent Probst, Jean-Jacques Schott, Julien Barc. ✉e-mail: Jean-Jacques.Schott@univ-nantes.fr; julien.barc@univ-nantes.fr

