## [Peer Review File · Nature Communications]

TAD boundary deletion causes PITX2-related cardiac electrical and structural defectsREVIEWER COMMENTS

Reviewer #1 (Remarks to the Author):

This study is both interesting and significant as it presents a new type of heart disease that is caused by the deletion of the 4q25 intergenic region. The deletion affects chromatin conformation and Pitx2 expression, leading to a range of complex cardiac phenotypes, such as sinoatrial node dysfunction and cardiac morphological abnormalities, as supported by a wealth of clinical case data. In addition, the authors utilized hiPSC-CMs and mouse models to conduct functional studies that demonstrate the pathogenic cause of such diseases is chromatin conformation remodeling, which leads to opposite regulation of Pitx2 between the ventricle and right atrium. This study successfully combines clinical and basic research to offer new insights into the genetic basis and molecular mechanisms of rare heart diseases linked to the 4q25 deletion. However, there are still some limitations for improvement, as follows:

1. If possible, please provide some simple explanations of the relevant technical terms used in the second part of the description of the cardiac function phenotype, to make it easier for readers from non-clinical fields.
2. Does the significant differential expression of some genes near the deleted region detected in different cardiac regions of mouse models contribute to some extent to the phenotype?
3. In the final mechanistic part, this study mainly used datasets from databases and ChIP data from hiPSC-CM WT/15kb-/- to find that regulatory elements in differential accessibility regions between ventricles and right atria are mainly responsible for opposite regulation of Pitx2 expression. These evidences support well previous findings. However, this part ends abruptly and seems a bit thin. Something in detail might be explored by using existing cell models. For example, how changes in chromatin conformation between the atria and ventricles are related to Pitx2 expression.
4. Please discuss whether there is any correlation between the deletion size or location and the phenotype severity or complexity among different cases. For example, if there is any association between the length of the deleted region, the degree of abnormal expression of Pitx2 gene and the severity of pathology?
5. TAD, loop and CTCF modification indeed affect cardiac morphology and function. The authors may discuss a recent paper published by Gong et al, Cell Regen, 2022.
6. In my opinion, if the authors could explain or discuss further how abnormal expression of Pitx2 regulates sinus node development and function at molecular level (e.g., involving which transcriptional targets or pathways) would be better.

Reviewer #2 (Remarks to the Author):

In the submitted manuscript by Baudic et al., the authors interrogate a series of overlapping 4q25 deletions identified in several families. Carriers had high incidence of sinus node dysfunction, increased RR intervals, and early onset atrial fibrillation as well as frequent occurrence of mitral valve prolapse, left-ventricular non-compaction, and atrial septal defects. A mouse model was generated using CRISPR/Cas9 to the corresponding human deletion, which demonstrated similar cardiac phenotypes, while adjacent deletions had no effect. These deletions in human induced pluripotent stem cells (hiPSCs) demonstrated conformational changes in higher order chromatin structure of the locus, bringing the PITX2 promoter into greater contact with the neighboring topologically associated domain (TAD) and cis-regulatory elements present within. The mutant mouse model and mutant hiPSC differentiated to ventricular cardiomyocytes both demonstrated increased expression in the right atrium and ventricular cardiomyocytes, locations with lower or no PITX2 expression. Overall, these observations are of high scientific value and provide greater insight into the 4q25 locus and PITX2 gene regulation related to atrial fibrillation.

Comments and concerns for the reviewers:

1. These 4q25 deletions are near the mouse lncRNA D030025E07Rik, sometimes referred to as Playrr in the literature. Previously (Welsh et al. Cell Reports 2015), Natasza Kurpios and her colleagues described an interaction between the Pitx2 promoter and a cis-regulatory element

downstream of the D030025E07Rik promoter, which based on homology should be in the “Non-coding TAD”, perhaps even corresponding to R1 or R2 (Figure 3a). This enhancer-promoter interaction was shown to be involved in left atrial expression of Pitx2 during development. On the other hand, expression of D030025E07Rik in the right atrium inhibited this 3D chromatin interaction and prevents right atrial Pitx2 expression. Given this previous observation, I have a couple interrelated questions:

a. HiC was performed in hiPSC-derived ventricular cardiomyocytes, a cell type that does not appear to express PITX2 appreciably. In this condition, there was two distinct TADs, an observation that would be predicted from the previous Kurpios data. If HiC was performed in genetically wildtype hiPSC-derived left atrial cardiomyocytes, would the “Fused-TAD” observation be the predicted norm?

b. Given the antagonistic nature of D030025E07Rik and Pitx2 expression in development, is there decreased expression of D030025E07Rik in the developing right atrium of DelB mice? What about the P21 right atrium of DelB mice?

2. The authors suggest that elements R1 and R2 may represent cis-regulatory elements that become engaged with the PITX2 promoter in the 4q25 deletions. Are those elements sufficient to drive expression in either hiPSC-derived atrial/ventricular cardiomyocytes or mouse cardiac tissue? Additionally, is there evidence that these elements are engaged in the left atrium, alluding back to the above question.

3. Was there any interaction between the previously described atrial fibrillation-associated enhancer region upstream of PITX2 and the Fused-TAD? Is there any evidence that the previously described SNPs within that AF-associated region impact the 3D chromatin and the described 4q25 deletions within this manuscript?

Reviewer #3 (Remarks to the Author):

In this work, Baudic and colleagues have described how the deletion of a CTCF boundary at the PITX2 locus induces a change in the chromatin organization which leads to the misexpression of multiple genes among them PITX2. Moreover, the authors show how this expression alteration observed, both in hiPSC-CMs and mice models with this deletion, can explain the phenotype observed in the patients of several families with this boundary deleted.

Overall this work is well structured and concise and contains data of high value, however, there are several minor issues that, in the opinion of this reviewer, should be addressed for a better understanding of the paper:

- Although the authors mention that they performed the orthologous 15kb deletion identified in humans in mice, there is a lack of explanation of how is the conservation of the locus between human and mice. This statement is necessary to understand if similar regulatory mechanisms can be shared between mice and humans. Indeed, it will be helpful to the reader to visualize, in the mouse genome, a CTCF track from mice heart to see the conservation of the deleted peak, especially since there is already published data available from 8 weeks mouse adult heart.

- One of the key milestones of the paper is that in the 15kb deletion present in the patients the authors identified a CTCF boundary that when is lost causes the fusion of PITX2-TAD and non-coding-TAD in one unique TAD. However, authors should show the orientation of the CTCF peaks identified within the locus of interest. This will provide a better explanation on how the fusion of TADs occurs upon CTCF site deletion.

- It is not clear to this reviewer if the hiPSC-CMs HiC, ATAC-seq, and CUT & RUN experiments (H3K27ac, H3K4me3, H3K4me1 and CTCF) have been done on hiPSC-CMs ventricular liked, sinoatrial node or from a differentiation that can contain a mix of different cardiac cell-types.

- In mouse hearts, Pitx2 is mainly expressed in the left atria and there is almost no reported expression of Pitx2 in the right atria in the literature. Baudic and colleagues observed in P21 mice with the 15kb deletion a clear gain of Pitx2 expression in the right atria and a decreased

expression in the ventricle. In parallel, they detected in hiPSC-CMs with the 15kb deletion a gain of PITX2 expression in SAN like cells and a loss of expression in the ventricular like cells. It will bring more value to the paper if the authors discuss further the relevance of this discovery and its links to the phenotype observed in patients in the discussion.

- In relation to this previous comment, authors explore if this tissue-specific gene expression alteration is due to the change in chromatin structure observed upon the loss of a CTCF site. It is not clear to this reviewer, when in line 158 they mention that they identified a new region interacting with PITX2, which region the authors refer to. Especially since it is difficult to determine if there is a gain or a loss with the R1 and R2 regions. It is also unclear if the authors are proposing these two regions as potential tissue specific regulatory regions, since it is not shown if there is also tissue specific H3K27ac. Moreover, there is also a lack of explanation whether the transcription factors identified also can contribute or not to this tissue specificity and therefore to the gene expression changes observed in hiPSC-CMs with the 15kb deletion. In summary, a larger discussion on the potential mechanism driving the changes in gene expression is needed.

- Finally, a broader introduction describing the locus together with the relevance of the main gene discussed across the manuscript, PITX2, needs to be included for an audience with no experience in this locus can understand the relevance of the question that the authors want to answer.

Other minor comments:

- It will be helpful to provide the precise coordinates of mice deletions in the text.

- In Figure 2, chromatin structure experiment is labeled as Hi-C meanwhile in Figure 3a is labeled as Capture Hi-C. It should be clarified whether all the experiments are Hi-C or Capture Hi-C since in the Material and Methods there is only one section referring to Capture Hi-C.

- Line 406-407 of Material and Methods "For tissue harvest, animals were euthanized by 20% CO₂ inhalation followed by cervical dislocation" is not related to the section 10 of Material and Methods "iPSC generation and cardiomyocyte ventricular like differentiation".

Thank you again for submitting your manuscript "TAD boundary deletion causes PITX2-related cardiac electrical and structural defects" to Nature Communications. We have now received reports from 3 reviewers and, on the basis of their comments, we have decided to invite a revision of your work for further consideration in our journal. Your revision should address all the points raised by our reviewers (see their reports below).

REVIEWER COMMENTS

Reviewer #1 (Remarks to the Author):

This study is both interesting and significant as it presents a new type of heart disease that is caused by the deletion of the 4q25 intergenic region. The deletion affects chromatin conformation and Pitx2 expression, leading to a range of complex cardiac phenotypes, such as sinoatrial node dysfunction and cardiac morphological abnormalities, as supported by a wealth of clinical case data. In addition, the authors utilized hiPSC-CMs and mouse models to conduct functional studies that demonstrate the pathogenic cause of such diseases is chromatin conformation remodeling, which leads to opposite regulation of Pitx2 between the ventricle and right atrium. This study successfully combines clinical and basic research to offer new insights into the genetic basis and molecular mechanisms of rare heart diseases linked to the 4q25 deletion. However, there are still some limitations for improvement, as follows:

1. If possible, please provide some simple explanations of the relevant technical terms used in the second part of the description of the cardiac function phenotype, to make it easier for readers from non-clinical fields.

We thank the reviewers for pointing this. We completed the phenotype description with a definition for non-clinical readers: Please see modification below and in the revised manuscript from row 89 to 107.

"Sinus node dysfunction, which is characterized by sinus bradycardia with sometimes cardiac pauses, was the main phenotype for 94% of the deletion carriers (n=65/69, mean age: 13.3 y.o. +/- 14.4), of whom 60% benefited from pacemaker implantation (n=39/65, mean age: 28 y.o. +/- 17.6) (Fig. 1c, Supplementary Table 1, Extended Data Fig. 1a). No sinus node dysfunction was identified among non-carriers (P<2.2E-16) (Fig. 1c). Heart rate measured by the RR interval (duration between two heart beats) was significantly longer in deletion carriers compared to non-carriers (1037+/-315 vs. 758+/-184 ms, P<1.0E-04). Of note, 45% of carriers (n=26/58) presented atrial fibrillation (mean age: 40 y.o. +/- 14.7) vs 3% of the non-carriers (P=2.1E-05) (Table 1, Fig. 1c). The ventricular repolarization duration (QTc), while remaining within the normal limits, was longer among subjects carrying the deletion than among non-carriers (mean QTc: 421+/-43 vs. 397+/-30 ms, P=0.02). Similarly, the ventricular repolarization duration including the ECG "U" wave QTcU was significantly prolonged in deletion carriers compared to non-carriers (575+/-79 vs. 441+/-51 ms, P<1.0E-04). Cardiac morphological abnormalities were found in 72% (n=42/58) of the clinically explored deletion carriers, whereas they were found in only 14% (n=4/28) of the individuals carrying no deletion (P=4.6E-07). These abnormalities included mitral valve prolapse/billowing (38% of deletion carriers vs 15% in non-carriers; P=4.3E-02), left-ventricular non-compaction (24% vs 0%; P=3.8E-03) and atrial septal defect (24% vs

0% $P=3.7E-03$) (Table 1, Fig. 1c, Extended Data Fig. 1b). Together, these electrical anomalies associated with cardiac structural defects constitute a new cardiac entity with a high penetrance (97%, $n=67/69$) (Table 1; Supplementary Table 1).”

2. Does the significant differential expression of some genes near the deleted region detected in different cardiac regions of mouse models contributes to some extent to the phenotype?

Thank you for this relevant question. To our knowledge, no mutations have been found in these genes in patients with a cardiac phenotype except for ANK2, the expression of which is not affected. Furthermore, enrichment for rare variants in these genes has not been associated with a cardiac phenotype in the UKBiobank database (<https://app.genebass.org>). RNA-seq of SAN tissue from WT and DelB mice (new data, see revised Figure 4) did not show dysregulation of genes within the surrounding 3Mb except for an RNA binding protein, Larp7. Larp7 is slightly upregulated in DelB SAN tissue and has been recently implicated in mitochondrial homeostasis and the maintenance of cardiac function. However, altered expression of these genes cannot be distinguished from the extensive transcriptomic remodeling induced by the deletion that causes ectopic PITX2 expression: Over 4000 genes are significantly deregulated in hiPSC-CM CTCF-/- model compared to isogenic controls (RNA-seq experiment described in methods and in extended Figure 5) and 2179 genes in the SAN tissue between WT and delB mice.

We provide new RNA-seq data from the mouse model and a comparison with publicly available single cell RNA-seq data to provide insight into the transcriptional response to ectopic Pitx2 expression. See Result section row from 158 to 178 and revised Figure 4.

“We separately microdissected SAN tissue and RA tissue without SAN of adult WT ($n=8$) and DelB ($n=7$) mice and performed whole-tissue RNA sequencing (Figure 4b-d). We observed that Pitx2 was significantly induced only in DelB SAN tissue (1.04 L2FC, $padj=1.26E-04$) and not in RA tissue (0.07 L2FC, $padj=0.58$). These data reveal that Pitx2 is specifically induced in SAN tissue of DelB mice. In SAN tissue, we found 2182 differentially expressed genes ($P_{adj}<0.05$; Fig.4c; supplementary table 2) of which 1003 were upregulated and 1179 were downregulated in DelB mice. We additionally found that the expression of 13 genes located within 3 Mb around the deletion was unchanged, with the exception of Larp7, an RNA binding protein recently implicated in mitochondrial homeostasis and the maintenance of cardiac function¹⁹, which was slightly upregulated in DelB SAN tissue (Figure 4b; Supplementary table 2). Gene Ontology (GO) analysis yielded terms for neuronal and development processes for genes downregulated in the DelB SAN and terms for metabolic processes for genes upregulated in the DelB SAN (Supplementary tables 3 and 4). We additionally noted that the ectopic expression of Pitx2 in the SAN region was accompanied by a marked reduction in pacemaker cell-associated gene expression in DelB SAN tissue, including key transcription factor genes (Tbx3, Isl1), ion channels (Hcn4, Cacna2d2, Ryr2) and other SAN markers (Vsnl1, Ntm, Cpne5)²⁰⁻²⁴. Tbx3 and Isl1 are essential for SAN development^{20,25,26} and Hcn4 encodes the hyperpolarization-activated cyclic nucleotide-gated K^+ channel that mediates the spontaneous activation of pacemaker cells in the SAN and has been implicated in SND²⁷. To further explore the changes in pacemaker/SAN-enriched gene expression in our whole-tissue RNA-seq data, we compared the genes differentially expressed in the DelB SAN with those previously found by single cell RNA-sequencing to be enriched or depleted in fetal pacemaker cardiomyocytes compared to atrial cardiomyocytes²²”

3. In the final mechanistic part, this study mainly used datasets from databases and cHi-C data from hiPSC-CM WT/15kb/- to find that regulatory elements in differential accessibility regions between ventricles and right atria are mainly responsible for opposite regulation of Pitx2 expression. These evidences support well previous findings. However, this part ends abruptly and seems a bit thin. Something in detail might be explored by using existing cell models. For example, how changes in chromatin conformation between the atria and ventricles are related to Pitx2 expression.

Thank you for the suggestions. Unfortunately, differentiation in atrial like cells or ventricular like cells present limitations such as to not produce “pure” atrial or ventricular cells but rather an enrichment in immature atrial or ventricular like cells. Data from these mixed populations of cardiomyocyte sub-types is difficult to interpret in terms of atrial or ventricular specificity. Publicly available HiC data from human atrial and ventricular tissue show that in both compartments the two (sub)TADs are separated (see Figure 1 below).

Figure 1. Hi-C data set from Ralf Gilsbach’s lab shows that both TADs are visible in purified cardiomyocyte nuclei from human left ventricular (hLV) and atrial tissue (hLA).

This suggests that in different cardiac cells TAD fusion can occur due to the deletion of the TAD boundary, as was observed in our iPSC-CM model. Furthermore, we noted that the ectopic expression

of PITX2 is limited to the SAN region within the right atrium (see new RNAseq data of mouse *DelB* SAN tissue and revised Figure 4. Publicly available ATAC-seq datasets (CARE database) show differences in chromatin accessibility between human atrial cardiomyocytes, ventricular tissue, and human pacemaker-like cardiomyocytes, indicating the presence of cell type-specific regulatory sequences. We propose that the deletion-induced conformational change in all cell types has different consequences for pacemaker cells (PITX2 induction), atrial cells (no change in PITX2 expression) and ventricular cells (PITX2 reduction) because in pacemaker cells, pacemaker-specific enhancers ectopically interact with the PITX2 promoter whereas in ventricular cells interaction with ventricular enhancers may be lost (or interactions with repressing sequences may be gained).

To dig into the characterization of these new regulatory regions interacting with PITX2, we further investigated the role of the R1 regulatory region that overlaps with the lncRNA *D030025E07Rik*, also known as *Playrr* in mouse. This lncRNA and *Pitx2* are expressed in a mutually exclusive manner in the developing gut, and mutual antagonism between *Pitx2* and *Playrr* was previously implicated in the left-right asymmetric morphogenesis of the gut (Welsh et al. Cell Reports 2015). We then investigated the expression pattern of *Playrr* in mouse and human SAN and right atrial tissue (see revised Figure 5) and in WT and *DelB* SAN tissue (see revised Figure 5). The data suggest that the *Pitx2-Playrr* mutual antagonism in gut may also be involved in *Pitx2* repression in the SAN. We extended the “New interactions between atrial and ventricular-specific regulatory regions and PITX2” section accordingly from row 219 to 231:

“Interestingly, R1 corresponds to the mouse orthologue lncRNA *D030025E07Rik*, also known as *Playrr* (Fig. 3a). In the developing dorsal mesentery, *Playrr* and *Pitx2* are expressed in a mutually exclusive manner and their reciprocal repression has been proposed to contribute to the establishment of left-right asymmetry during gut looping¹⁸. Mouse *Playrr* is localized in the non-coding TAD. We interrogated SAN and RA-specific RNA-seq data sets²⁰ and found that *Playrr* is expressed in the early fetal SAN and to a lesser extent in the RA (P=0.02) (Fig. 5a). Just before birth, the levels of *Playrr* expression were similar in SAN and RA and after birth, *Playrr* expression was decreased in both tissues (Fig.5b). Although *PLAYRR* has not been annotated in the human genome, we observed SAN-enriched expression of a human orthologous transcript (*PLAYRR*) in RNA-seq data from fetal human SAN and RA²¹ (Fig. 5c). In both species, the transcript is initiated from a conserved promoter and appears to be poorly spliced. RNA sequencing of the adult micro-dissected SANs reveals a marked reduction in *Playrr* expression in *DelB* mice as compared to WT mice (L2FC=-2.45, P = 0.007), complementing the SAN-specific gain in *Pitx2* expression (L2FC=2.14, P = 0.0002) (Fig. 5d).”

4. Please discuss whether there is any correlation between the deletion size or location and the phenotype severity or complexity among different cases. For example, if there is any association between the length of the deleted region, the degree of abnormal expression of *Pitx2* gene and the severity of pathology?

Thank you for this relevant comment. We could not find striking differences between families despite differences in deletion size. This is in line with the fact that deletions upstream or downstream of the 15Kb deletion in the mouse models (DeIA and DeIC) did not cause a phenotype. Unfortunately, expression of PITX2 in relation to deletion size in human cells has not been tested, as we only generated an iPSC model with the 15Kb deletion.

5. TAD, loop and CTCF modification indeed affect cardiac morphology and function. The authors may discuss a recent paper published by Gong et al, Cell Regen, 2022.

We agree with the reviewer that previous studies have demonstrated the pivotal role of TAD boundaries in gene regulation and the potential impact on phenotype in case of deletion of these boundaries (Lupiáñez DG. et al. 2015, and Rajderkar, S. et al. 2023). More broadly, point mutation of the CTCF transcription factor like in Gong et al, Cell Regen, 2022 seems to affect the mesoderm differentiation of embryonic stem cells. More interestingly, its specific depletion in a cardiac-specific model has been demonstrated by Rosa-Garrido, M. et al. 2017 leading to heart failure.

We added the following sentences to the introduction accordingly:

“Genome editing of TAD boundaries in mice leads to aberrant gene regulation and altered phenotypes.⁶ In the mammalian genome, CTCF is dose-dependently required for looping between CTCF target sites and insulation of topologically associating domains (TADs).⁷ In a mouse model, cardiomyocyte-specific inducible depletion of CTCF caused profound transcriptional dysregulation and heart failure.⁸”

6. In my opinion, if the authors could explain or discuss further how abnormal expression of Pitx2 regulates sinus node development and function at molecular level (e.g., involving which transcriptional targets or pathways) would be better.

We performed RNA-seq experiments on WT and DelB mouse SAN tissue, which uncovered pronounced transcriptional changes as a consequence of Pitx2 dysregulation. We specifically highlighted the expression remodeling of key transcription factor genes (Tbx3, Isl1), ion channels (Hcn4, Cacna2d2, Ryr2) and other SAN-specific markers (Vsnl1, Ntm, Cpne5). The data is presented in revised Figure 4 and in the results section from row 158 to178:

*“We separately microdissected SAN tissue and RA tissue without SAN of adult WT (n=8) and DelB (n=7) mice and performed whole-tissue RNA sequencing (Figure 4b-d). We observed that Pitx2 was significantly induced only in DelB SAN tissue (1.04 L2FC, padj=1.26E-04) and not in RA tissue (0.07 L2FC, padj=0.58). These data reveal that Pitx2 is specifically induced in SAN tissue of DelB mice. In SAN tissue, we found 2182 differentially expressed genes ($P_{adj} < 0.05$; Fig.4c; supplementary table 2) of which 1003 were upregulated and 1179 were downregulated in DelB mice. We additionally found that the expression of 13 genes located within 3 Mb around the deletion was unchanged, with the exception of *Larp7*, an RNA binding protein recently implicated in mitochondrial homeostasis and the maintenance of cardiac function¹⁹, which was slightly upregulated in DelB SAN tissue (Figure 4b; Supplementary table 2). Gene Ontology (GO) analysis yielded terms for neuronal and development processes for genes downregulated in the DelB SAN and terms for metabolic processes for genes upregulated in the DelB SAN (Supplementary tables 3 and 4). We additionally noted that the ectopic expression of Pitx2 in the SAN region was accompanied by a marked reduction in pacemaker cell-associated gene expression in DelB SAN tissue, including key transcription factor genes (Tbx3, Isl1), ion channels (Hcn4, Cacna2d2, Ryr2) and other SAN markers (Vsnl1, Ntm, Cpne5)²⁰⁻²⁴. Tbx3 and Isl1 are essential for SAN development^{20,25,26} and Hcn4 encodes the hyperpolarization-activated cyclic nucleotide-gated K⁺ channel that mediates the spontaneous activation of pacemaker cells in the SAN and has been implicated in SND²⁷. To further explore the changes in pacemaker/SAN-enriched gene expression in our*

whole-tissue RNA-seq data, we compared the genes differentially expressed in the DelB SAN with those previously found by single cell RNA-sequencing to be enriched or depleted in fetal pacemaker cardiomyocytes compared to atrial cardiomyocytes²²

And in the discussion section from 274 to 297:

“Conversely, here we see the severe disruption of the SAN-specific genetic program in the right-sided DelB SAN region coinciding with the induction of ectopic Pitx2 expression in the SAN. Among the suppressed genes in DelB SAN tissue are genes encoding the key pacemaker transcription factors Tbx3 and Isl1 and ion channels that drive pacemaker function including Hcn4. Both Tbx3 and Isl1 activate and maintain the pacemaker genetic program in the heart, preventing the expression of genes associated with the working atrial myocardium in the SAN domain and their loss compromises SAN development, culminating in SAN dysfunction^{20,21,25,26}. Cardiac-specific ablation of Hcn4 expression in mice results in severe and lethal bradycardia^{39,40}. We confirmed the loss of the pacemaker genetic program in the DelB SAN by comparing differentially expressed genes with those enriched in pacemaker cardiomyocytes²². We found additional gain of atrium-specific markers in the DelB SAN tissue. The loss of SAN-associated gene expression, as indicated by a marked reduction in Isl1 and Tbx3 expression, in adult DelB mice indicates that Pitx2 is not only necessary, but also sufficient to suppress the SAN genetic program.

Tbx5 and Pitx2 form an incoherent feed-forward loop that is driven by Tbx5 and suppressed by Pitx2, governing a left atrial transcriptional network that regulates rhythm effector gene expression⁴¹. While reduced Tbx5 expression in adult mice disrupts the expression of calcium handling and AF-susceptibility genes, reducing myocardial automaticity and elevating atrial fibrillation susceptibility, these phenotypes can be rescued by Pitx2 haploinsufficiency. While Tbx5 expression seems unaffected in the DelB SAN, the Tbx5-Pitx2 balance may be disrupted in the SAN due to the ectopic expression of Pitx2. We see that some, but not all, previously identified targets of the Tbx5-Pitx2 regulatory loop, including those associated with calcium handling (Ryr2, Pln) and rapid depolarization (Scn5a) are differentially expressed in the DelB SAN, potentially driving the susceptibility to atrial arrhythmias and SAN dysfunction in DelB mice.”

Reviewer #2 (Remarks to the Author):

In the submitted manuscript by Baudic et al., the authors interrogate a series of overlapping 4q25 deletions identified in several families. Carriers had high incidence of sinus node dysfunction, increased RR intervals, and early onset atrial fibrillation as well as frequent occurrence of mitral valve prolapse, left-ventricular non-compaction, and atrial septal defects. A mouse model was generated using CRISPR/Cas9 to the corresponding human deletion, which demonstrated similar cardiac phenotypes, while adjacent deletions had no effect. These deletions in human induced pluripotent stem cells (hiPSCs) demonstrated conformational changes in higher order chromatin structure of the locus, bringing the PITX2 promoter into greater contact with the neighboring topologically associated domain (TAD) and cis-regulatory elements present within. The mutant mouse model and mutant hiPSC differentiated to ventricular cardiomyocytes both demonstrated increased expression in the right atrium and ventricular cardiomyocytes, locations with lower or no PITX2 expression. Overall, these observations are of high scientific value and provide greater insight into the 4q25 locus and PITX2 gene regulation related to atrial fibrillation.

Comments and concerns for the reviewers:

1. These 4q25 deletions are near the mouse lncRNA D030025E07Rik, sometimes referred to as Playrr in the literature. Previously (Welsh et al. Cell Reports 2015), Natasza Kurpios and her colleagues described an interaction between the Pitx2 promoter and a cis-regulatory element downstream of the D030025E07Rik promoter, which based on homology should be in the “Non-coding TAD”, perhaps even corresponding to R1 or R2 (Figure 3a). This enhancer-promoter interaction was shown to be involved in left atrial expression of Pitx2 during development. On the other hand, expression of D030025E07Rik in the right atrium inhibited this 3D chromatin interaction and prevents right atrial Pitx2 expression. Given this previous observation, I have a couple interrelated questions:

a. HiC was performed in hiPSC-derived ventricular cardiomyocytes, a cell type that does not appear to express PITX2 appreciably. In this condition, there was two distinct TADs, an observation that would be predicted from the previous Kurpios data. If HiC was performed in genetically wildtype hiPSC-derived left atrial cardiomyocytes, would the “Fused-TAD” observation be the predicted norm?

We thank the reviewer for this interesting comment. Since differentiation toward pacemaker, atrial or ventricular like cells from iPSC leads to only an enrichment of one of these cardiac cell types, the signals from HiC assays will always contains a mix of different cardiac compartment like cells. However, we investigated the chromatin architecture in human atrial and ventricular tissues from Ralf Gilsbach’s lab (see Figure 1 above: Reviewer#1’s section: purified cardiomyocyte nuclei from human left ventricular and atrial tissue) and from available datasets in ENCODE (Wouter de Laat’s lab): “PITX2” and the “non-coding” TADs can be distinguished in left ventricles as well as in left atria (see Figure 2 below).

b. Given the antagonistic nature of D030025E07Rik and Pitx2 expression in development, is there decreased expression of D030025E07Rik in the developing right atrium of DelB mice? What about the P21 right atrium of DelB mice?

We thank the reviewer for bringing this up. Firstly, we performed RNA-seq on adult WT and DelB microdissected RA and SAN tissue and observed that the ectopic expression of Pitx2 in the DelB whole right atrium (including SAN, original Figure 3) was in fact limited to SAN tissue (see revised Figure 4.) Secondly, we describe the expression pattern of D030025E07Rik/Playrr in the pre- and postnatal SAN and RA of mice using publicly available RNA-seq data (Vedantham et al. 2015), and in addition observed expression of the human orthologous lncRNA in fetal SAN RNA-seq data (Van Eif et al. 2019) (see revised Figure 5). Finally, we found a significant decrease in Playrr expression in DelB SAN tissue correlating with an increased expression of Pitx2 (see revised Figure 5). Accordingly, we extended the “New interactions between atrial and ventricular-specific regulatory regions and PITX2” section. Please see row 219 to 231 in the revised manuscript:

“Interestingly, R1 corresponds to the mouse orthologue lncRNA D030025E07Rik, also known as Playrr (Fig. 3a). In the developing dorsal mesentery, Playrr and Pitx2 are expressed in a mutually exclusive manner and their reciprocal repression has been proposed to contribute to the establishment of left-right asymmetry during gut looping¹⁸. Mouse Playrr is localized in the non-coding TAD. We interrogated SAN and RA-specific RNA-seq data sets²⁰ and found that Playrr is expressed in the early fetal SAN and to a lesser extent in the RA ($P=0.02$) (Fig. 5a). Just before birth, the levels of Playrr expression were similar in SAN and RA and after birth, Playrr expression was decreased in both tissues (Fig.5b). Although PLAYRR has not been annotated in the human genome, we observed SAN-enriched expression of a human orthologous transcript (PLAYRR) in RNA-seq data from fetal human SAN and RA²¹ (Fig. 5c). In both species, the transcript is initiated from a conserved promoter and appears to be poorly spliced. RNA sequencing of the adult micro-dissected SANs reveals a marked reduction in Playrr expression in DelB mice as compared to WT mice ($L2FC=-2.45$, $P = 0.007$), complementing the SAN-specific gain in Pitx2 expression ($L2FC=2.14$, $P = 0.0002$) (Fig. 5d).”

2. The authors suggest that elements R1 and R2 may represent cis-regulatory elements that become engaged with the PITX2 promoter in the 4q25 deletions. Are those elements sufficient to drive expression in either hiPSC-derived atrial/ventricular cardiomyocytes or mouse cardiac tissue? Additionally, is there evidence that these elements are engaged in the left atrium, alluding back to the above question.

We specifically interrogated ATAC-seq data from human iPSC-derived pacemaker cells (Van Eif et al. 2020) and observed that R1 and R2 harbor pacemaker-specific accessible regions in the non-coding TAD newly interacting with the PITX2 promoter. We also observed other pacemaker-specific accessible regions. Several of these regions also contain transcription factor binding sites involved in heart and SAN development. We propose that these candidate regulatory elements that interact with the PITX2 promoter in deletion carriers may contribute to the ectopic activation of PITX2 in SAN tissue. Accordingly, we have added these observations to the results section and the localization of these regions in the genome to revised Figure 3. and line 214-218.

“Interestingly, R1 and R2 harbor hiPSC-derived pacemaker cardiomyocyte-specific accessible chromatin regions (Fig. 3b)²⁸. Furthermore, binding sites for cardiac transcription factors^{29,30} such as ISL1, TBX5, GATA4, NKX2-5, MEIS1, TEAD and TBX5 co-localize with these accessible regions, supporting their potential role in cardiac gene expression regulation (Fig. 3b).

3. Was there any interaction between the previously described atrial fibrillation-associated enhancer region upstream of PITX2 and the Fused-TAD? Is there any evidence that the previously described SNPs within that AF-associated region impact the 3D chromatin and the described 4q25 deletions within this manuscript?

This is a relevant point, however, the resolution of the HiC data was not sufficient to clearly identify gain or loss of interactions between the AF-associated region and the PITX2 promoter or other sites. The AF-associated region is located near PITX2 and at more than 500Kb from the TAD boundary that is deleted in patients and the mouse model, suggesting this region may not be highly involved in the changed chromatin conformation in deletion carriers.

Reviewer #3 (Remarks to the Author):

In this work, Baudic and colleagues have described how the deletion of a CTCF boundary at the PITX2 locus induces a change in the chromatin organization which leads to the misexpression of multiple genes among them PITX2. Moreover, the authors show how this expression alteration observed, both in hiPSC-CMs and mice models with this deletion, can explain the phenotype observed in the patients of several families with this boundary deleted.

Overall this work is well structured and concise and contains data of high value, however, there are several minor issues that, in the opinion of this reviewer, should be addressed for a better understanding of the paper:

- Although the authors mention that they performed the orthologous 15kb deletion identified in humans in mice, there is a lack of explanation of how is the conservation of the locus between human and mice. This statement is necessary to understand if similar regulatory mechanisms can be shared between mice and humans. Indeed, it will be helpful to the reader to visualize, in the mouse genome, a CTCF track from mice heart to see the conservation of the deleted peak, especially since there is already published data available from 8 weeks mouse adult heart.

We thank the reviewer for this comment. We provided a Supplementary Figure 1 corresponding to an UCSC browser screenshots of the diverging CTCF binding sites deleted in the corresponding mouse and human genome.

- One of the key milestones of the paper is that in the 15kb deletion present in the patients the authors identified a CTCF boundary that when is lost causes the fusion of PITX2-TAD and non-coding-TAD in one unique TAD. However, authors should show the orientation of the CTCF peaks identified within the locus of interest. This will provide a better explanation on how the fusion of TADs occurs upon CTCF site deletion.

Thank you for this comment. We have added arrows showing the orientation of the CTCF sites in the revised Figure 2 and in Supplementary Figure 1.

- It is not clear to this reviewer if the hiPSC-CMs Hi-C, ATAC-seq, and CUT & RUN experiments (H3K27ac, H3K4me3, H3K4me1 and CTCF) have been done on hiPSC-CMs ventricular like, sinoatrial node or from a differentiation that can contain a mix of different cardiac cell-types.

The Hi-C, ATAC-seq and CUT&RUN experiments have been done using a differentiation protocol to obtain ventricular-like CMs from hiPSCs. However, despite the large numbers of ventricular-like CMs, other cardiomyocyte types (atrial, embryonic outflow tract, etc.) are also present. The ENCODE Registry of candidate cis-Regulatory Elements (cCREs) annotates this region as a “CTCF-only” region. Neither promoter nor enhancer signature can be detected among the annotated cCREs.

- In mouse hearts, Pitx2 is mainly expressed in the left atria and there is almost no reported expression of Pitx2 in the right atria in the literature. Baudic and colleagues observed in P21 mice with the 15kb deletion a clear gain of Pitx2 expression in the right atria and a decreased expression in the ventricle. In parallel, they detected in hiPSC-CMs with the 15kb deletion a gain of PITX2 expression in SAN like cells and a loss of expression in the ventricular like cells. It will bring more value to the paper if the authors discuss further the relevance of this discovery and its links to the phenotype observed in patients in the discussion.

We have performed additional transcriptional profiling of the mouse model (see revised Figure 4). These analyses revealed Pitx2 is selectively induced in the sinoatrial node (SAN) region/pacemaker cells within the right atrium (RA). These data provide an explanation for the sinus node dysfunction in patients. We added the following text to the results section row 158 to 178:

*“We separately microdissected SAN tissue and RA tissue without SAN of adult WT (n=8) and DelB (n=7) mice and performed whole-tissue RNA sequencing (Figure 4b-d). We observed that Pitx2 was significantly induced only in DelB SAN tissue (1.04 L2FC, padj=1.26E-04) and not in RA tissue (0.07 L2FC, padj=0.58). These data reveal that Pitx2 is specifically induced in SAN tissue of DelB mice. In SAN tissue, we found 2182 differentially expressed genes ($P_{adj} < 0.05$; Fig.4c; supplementary table 2) of which 1003 were upregulated and 1179 were downregulated in DelB mice. We additionally found that the expression of 13 genes located within 3 Mb around the deletion was unchanged, with the exception of *Larp7*, an RNA binding protein recently implicated in mitochondrial homeostasis and the maintenance of cardiac function¹⁹, which was slightly upregulated in DelB SAN tissue (Figure 4b; Supplementary table 2). Gene Ontology (GO) analysis yielded terms for neuronal and development processes for genes downregulated in the DelB SAN and terms for metabolic processes for genes upregulated in the DelB SAN (Supplementary tables 3 and 4). We additionally noted that the ectopic expression of Pitx2 in the SAN region was accompanied by a marked reduction in pacemaker cell-associated gene expression in DelB SAN tissue, including key transcription factor genes (*Tbx3*, *Isl1*), ion channels (*Hcn4*, *Cacna2d2*, *Ryr2*) and other SAN markers (*Vsnl1*, *Ntm*, *Cpne5*)²⁰⁻²⁴. *Tbx3* and *Isl1* are essential for SAN development^{20,25,26} and *Hcn4* encodes the hyperpolarization-activated cyclic nucleotide-gated K^+ channel that mediates the spontaneous activation of pacemaker cells in the SAN and has been implicated in SND²⁷. To further explore the changes in pacemaker/SAN-enriched gene expression in our whole-tissue RNA-seq data, we compared the genes differentially expressed in the DelB SAN with those previously found by single cell RNA-sequencing to be enriched or depleted in fetal pacemaker cardiomyocytes compared to atrial cardiomyocytes²²”*

And in the discussion section from 274 to 297:

“Conversely, here we see the severe disruption of the SAN-specific genetic program in the right-sided DelB SAN region coinciding with the induction of ectopic Pitx2 expression in the SAN. Among the suppressed genes in DelB SAN tissue are genes encoding the key pacemaker transcription factors Tbx3 and Isl1 and ion channels that drive pacemaker function including Hcn4. Both Tbx3 and Isl1 activate and maintain the pacemaker genetic program in the heart, preventing the expression of genes associated with the working atrial myocardium in the SAN domain and their loss compromises SAN development, culminating in SAN dysfunction^{20,21,25,26}. Cardiac-specific ablation of Hcn4 expression in mice results in severe and lethal bradycardia^{39,40}. We confirmed the loss of the pacemaker genetic program in the DelB SAN by comparing differentially expressed genes with those enriched in pacemaker cardiomyocytes²². We found additional gain of atrium-specific markers in the DelB SAN tissue. The loss of SAN-associated gene expression, as indicated by a marked reduction in Isl1 and Tbx3 expression, in adult DelB mice indicates that Pitx2 is not only necessary, but also sufficient to suppress the SAN genetic program.

Tbx5 and Pitx2 form an incoherent feed-forward loop that is driven by Tbx5 and suppressed by Pitx2, governing a left atrial transcriptional network that regulates rhythm effector gene expression⁴¹. While reduced Tbx5 expression in adult mice disrupts the expression of calcium handling and AF-susceptibility genes, reducing myocardial automaticity and elevating atrial fibrillation susceptibility, these phenotypes can be rescued by Pitx2 haploinsufficiency. While Tbx5 expression seems unaffected in the DelB SAN, the Tbx5-Pitx2 balance may be disrupted in the SAN due to the ectopic expression of Pitx2. We see that some, but not all, previously identified targets of the Tbx5-Pitx2 regulatory loop, including those associated with calcium handling (Ryr2, Pln) and rapid depolarization (Scn5a) are differentially expressed in the DelB SAN, potentially driving the susceptibility to atrial arrhythmias and SAN dysfunction in DelB mice.”

- In relation to this previous comment, authors explore if this tissue-specific gene expression alteration is due to the change in chromatin structure observed upon the loss of a CTCF site. It is not clear to this reviewer, when in line 158 they mention that they identified a new region interacting with PITX2, which region the authors refer to. Especially since it is difficult to determine if there is a gain or a loss with the R1 and R2 regions. It is also unclear if the authors are proposing these two regions as potential tissue specific regulatory regions, since it is not shown if there is also tissue specific H3K27ac. Moreover, there is also a lack of explanation whether the transcription factors identified also can contribute or not to this tissue specificity and therefore to the gene expression changes observed in hiPSC-CMs with the 15kb deletion. In summary, a larger discussion on the potential mechanism driving the changes in gene expression is needed.

We agree with the reviewer that this section deserved more attention. We extended the molecular investigations and the results section accordingly:

Please see row 219 to 231 in the revised manuscript:

“Interestingly, R1 corresponds to the mouse orthologue lncRNA D030025E07Rik, also known as Playrr (Fig. 3a). In the developing dorsal mesentery, Playrr and Pitx2 are expressed in a mutually exclusive manner and their reciprocal repression has been proposed to contribute to the establishment of left-

right asymmetry during gut looping¹⁸. Mouse *Playrr* is localized in the non-coding TAD. We interrogated SAN and RA-specific RNA-seq data sets²⁰ and found that *Playrr* is expressed in the early fetal SAN and to a lesser extent in the RA ($P=0.02$) (Fig. 5a). Just before birth, the levels of *Playrr* expression were similar in SAN and RA and after birth, *Playrr* expression was decreased in both tissues (Fig.5b). Although *PLAYRR* has not been annotated in the human genome, we observed SAN-enriched expression of a human orthologous transcript (*PLAYRR*) in RNA-seq data from fetal human SAN and RA²¹ (Fig. 5c). In both species, the transcript is initiated from a conserved promoter and appears to be poorly spliced. RNA sequencing of the adult micro-dissected SANs reveals a marked reduction in *Playrr* expression in *DelB* mice as compared to WT mice ($L2FC=-2.45$, $P = 0.007$), complementing the SAN-specific gain in *Pitx2* expression ($L2FC=2.14$, $P = 0.0002$) (Fig. 5d)."

and the discussion section as well (row 306-324):

"Non-coding regions enriched for epigenetic signatures associated with gene regulation predict the location of cis-regulatory elements (RE) that drive tissue- and context-specific gene expression⁴⁵. While REs and their target genes are often confined to the same TAD, their tissue-specific functional range can span entire gene deserts^{2,46,47}. For example, ablation of a regulatory element 1.1 Mb distal from *Tbx3* abolishes *Tbx3* expression in the SAN while its expression in other tissues is preserved⁴⁸. We have interrogated the regulatory landscape of the chr4q25 locus in ventricular, atrial and hiPSC-derived pacemaker-like cardiomyocytes and found a number of candidate pacemaker-specific REs that may mediate the ectopic activation of *PITX2* in the SAN region. Furthermore, we observed that the mutually exclusive expression of the lncRNA *Playrr*, located in the noncoding TAD, and *Pitx2*, previously described in the developing dorsal mesentery and implicated in the left-right asymmetry of the gut¹⁸, is mirrored in the pre- and postnatal mouse heart. Moreover, we identified expression of a human lncRNA, which based on conservation of location, promoter sequence and expression pattern is likely the human orthologue of *Playrr*, in the fetal human SAN. We noted a marked reduction in *Playrr* expression in the micro-dissected SAN of adult *DelB* mice, complimenting the SAN-specific gain in *Pitx2* expression. This observation indicates the *Pitx2-Playrr* mutual repression mechanism described in gut development¹⁸ may also be involved in *Pitx2* suppression in the SAN. We suggest that the loss of a boundary between the *PITX2*-TAD and the noncoding RNA-TAD permits ectopic interactions between the initially spatially separated *Pitx2* promoter and SAN-specific regulatory elements and/or *Playrr*, leading to ectopic activation of *Pitx2* expression in the SAN."

- Finally, a broader introduction describing the locus together with the relevance of the main gene discussed across the manuscript, *PITX2*, needs to be included for an audience with no experience in this locus can understand the relevance of the question that the authors want to answer.

We agree with the reviewer and extended the introduction on the potential role of TAD boundaries and associated CTCF binding sites in disease susceptibility as well as on the role of *PITX2* and its associated diseases (row 50-67).

"The formation and maintenance of loops and TADs involves the interplay of the cohesin complex and the DNA-binding protein CTCF, which bring distal sequences together⁵. Genome editing of TAD boundaries in mice leads to aberrant gene regulation and altered phenotypes.⁶ In the mammalian genome, CTCF is dose-dependently required for looping between CTCF target sites and insulation of

topologically associating domains (TADs).⁷ In a mouse model, cardiomyocyte-specific inducible depletion of CTCF caused profound transcriptional dysregulation and heart failure.⁸ These studies demonstrate the potential pathogenicity of genetic variants affecting CTCF sites at TAD boundaries. Here, we describe 7 families that share overlapping deletions in a 1.5 Mbp gene desert on chromosome 4q25 presenting with a novel cardiac entity including sinoatrial node (SAN) dysfunction, atrial fibrillation and developmental defects. The smallest deletion harbors two diverging CTCF binding sites and disrupts a TAD boundary. Among the genes located in the TADs, PITX2, a homeobox transcription factor, plays a key role in left-right asymmetrical organ development, including gut, stomach and heart, in cell fate determination, differentiation and organogenesis.⁹ Moreover, PITX2 is involved in regulation of genes underlying electrophysiological properties of the left atrium^{10,11} and has been implicated in atrial fibrillation by genome-wide association studies.¹² Pitx2 insufficiency in mice led to arrhythmogenesis and ectopic activation of aspects of the SAN genetic program in left atria¹³

Other minor comments:

- It will be helpful to provide the precise coordinates of mice deletions in the text.

We added in the Material and Methods “Generation of mutant mice” the following sentence: “Breakpoints for the DelA, DelB and DelC mouse models were: chr3:128,346,349-128,357,670, chr3:128,329,001-128,346,349 and chr3:128,270,995-128,328,982, respectively.”

- In Figure 2, chromatin structure experiment is labeled as Hi-C meanwhile in Figure 3a is labeled as Capture Hi-C. It should be clarified whether all the experiments are Hi-C or Capture Hi-C since in the Material and Methods there is only one section referring to Capture Hi-C.

We apologize for this mistake. We have performed only Hi-C (in situ Hi-C) in this manuscript.

- Line 406-407 of Material and Methods “For tissue harvest, animals were euthanized by 20% CO2 inhalation followed by cervical dislocation” is not related to the section 10 of Material and Methods “iPSC generation and cardiomyocyte ventricular like differentiation”.

Thank you for this comment, the sentence has been removed from the manuscript.

REVIEWERS' COMMENTS

Reviewer #2 (Remarks to the Author):

The authors have satisfactorily addressed all of the comments and questions raised by the other reviewers and myself. I appreciate the new transcriptional profiling data in the mutant mouse models and the attention to detail with the new line of investigation into the PITX2-antagonistic lncRNA. I have no further comments or questions.

Reviewer #3 (Remarks to the Author):

Baudic and colleagues have made a significant effort to address all the comments from the reviewers. Overall, the clarifications asked by this reviewer have been answered in a satisfactory manner. Thus, this revised version of the manuscript provides convincing data that support their conclusions. Only two minor points remain unclear, which will improve the manuscript further. This study provides very valuable insight into the regulatory mechanisms that control PITX2 locus and its association with atrial fibrillation and other cardiac diseases.

Minor comments:

It is unclear, to this reviewer, if the ATAC-seq from hiPSC-CMs shown in Figure 2 is the same as the one used in Figure 3b. In Figure 2 the peak of open chromatin at the TAD boundary (blue line highlight) is clear, meanwhile, this observation can't be made in Figure 3b for the track with the same name. In the opinion of this reviewer, an explanation for this difference should be provided.

It would help to understand better how the PITX2-TAD and the non-coding TAD are formed to know the directionality of all the CTCFs located within these two TADs. To know how the conserved and convergent CTCF sites identified at the TAD boundary located within the two TADs contact other CTCFs to form the mentioned TADS. However, this comment is more of a refinement comment rather than an issue.

We thank the reviewers for their positive comments and feedbacks.
Please find the answers of their minor comments below:

REVIEWER COMMENTS

It is unclear, to this reviewer, if the ATAC-seq from hiPSC-CMs shown in Figure 2 is the same as the one used in Figure 3b. In Figure 2 the peak of open chromatin at the TAD boundary (blue line highlight) is clear, meanwhile, this observation can't be made in Figure 3b for the track with the same name. In the opinion of this reviewer, an explanation for this difference should be provided.

We thank the reviewer for pointing this. We had noticed that peaks may disappear when we zoom in or out due to failure of display of the UCSC browser. We had used another track made of 2 different differentiation of the same clone to increase signal (brown track visible on Figure 2). We now updated Figure 3 with the same track.

It would help to understand better how the PITX2-TAD and the non-coding TAD are formed to know the directionally of all the CTCFs located within these two TADs. To know how the conserved and convergent CTCF sites identified at the TAD boundary located within the two TADs contact other CTCFs to form the mentioned TADS. However, this comment is more of a refinement comment rather than an issue.

We thank the reviewer for this suggestion. This is indeed a more fundamental aspect of the regulation of the PITX2 TAD and its environment and would deserve more experiments to have a comprehensive overview of the CTCF organization and function within this loci. The tight regulation of *PITX2* and its TAD organization may constitute a follow-up manuscript.